# Large-batch Optimization for Dense Visual Predictions: Training Faster R-CNN in 4.2 Minutes

**Zeyue Xue**[*]
The University of Hong Kong
xuezeyue@connect.hku.hk

**Jianming Liang**[*]
Beihang University
ljmmm1997@gmail.com

**Guanglu Song**
Sensetime Research
songguanglu@sensetime.com

**Zhuofan Zong**[*]
Beihang Univerisity
zongzhuofan@gmail.com

**Liang Chen**[*]
Peking University
clandzyy@pku.edu.cn

**Yu Liu**[†]
Sensetime Research
liuyuisanai@gmail.com

**Ping Luo**[†]
The University of Hong Kong,
Shanghai AI Laboratory
pluo@cs.hku.hk

## Abstract

Training a large-scale deep neural network in a large-scale dataset is challenging and time-consuming. The recent breakthrough of large-batch optimization is a promising way to tackle this challenge. However, although the current advanced algorithms such as LARS and LAMB succeed in classification models, the complicated pipelines of dense visual predictions such as object detection and segmentation still suffer from the heavy performance drop in the large-batch training regime. To address this challenge, we propose a simple yet effective algorithm, named Adaptive Gradient Variance Modulator (AGVM), which can train dense visual predictors with very large batch size, enabling several benefits more appealing than prior arts. Firstly, AGVM can align the gradient variances between different modules in the dense visual predictors, such as backbone, feature pyramid network (FPN), detection, and segmentation heads. We show that training with a large batch size can fail with the gradient variances misaligned among them, which is a phenomenon primarily overlooked in previous work. Secondly, AGVM is a plug-and-play module that generalizes well to many different architectures (*e.g.,* CNNs and Transformers) and different tasks (*e.g.,* object detection, instance segmentation, semantic segmentation, and panoptic segmentation). It is also compatible with different optimizers (*e.g.,* SGD and AdamW). Thirdly, a theoretical analysis of AGVM is provided. Extensive experiments on the COCO and ADE20K datasets demonstrate the superiority of AGVM. For example, it can train Faster R-CNN+ResNet50 in 4.2 minutes without losing performance. AGVM demonstrates more stable generalization performance than prior arts under extremely large batch size (*i.e.,* 10k). It enables training an object detector with one billion parameters in just 3.5 hours, reducing the training time by 20.9×, whilst achieving 62.2 mAP on COCO. The deliverables are released at https://github.com/Sense-X/AGVM.

---

[*]Work done during an internship at Sensetime Research.
[†]Corresponding authors.

36th Conference on Neural Information Processing Systems (NeurIPS 2022).

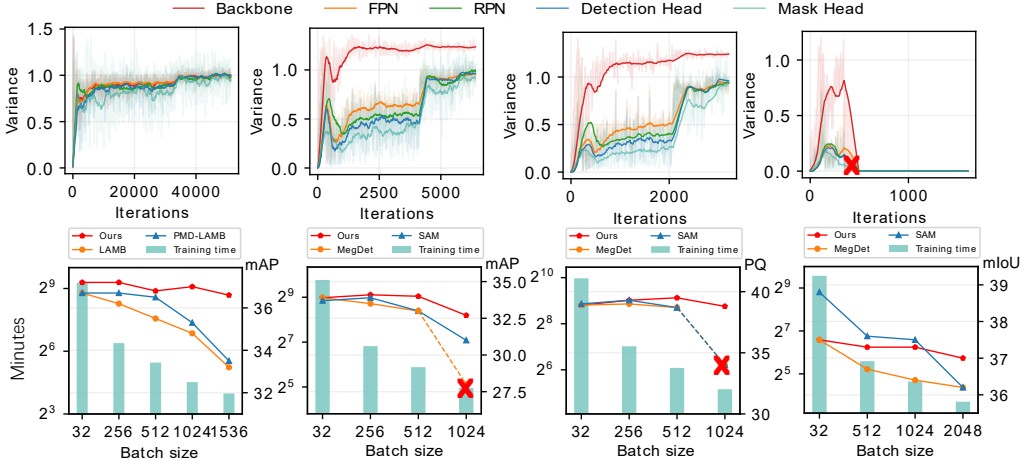

Figure 1: **First row**: Comparisons of the gradient variances (omitting learning rate in $\Phi_t^{(i)}$ referred to Eq. (3)) of different network modules in Mask R-CNN, including backbone, FPN, RPN, and heads. From left to right, the models are trained using SGD with a mini-batch size of 32, 256, 512, and 1024, respectively. Note that smaller batch size (32 in the first figure) produces similar $\Phi_t^{(i)}$ between different modules. When the batch size increases from 256 to 1024 ($2^{nd} \sim 4^{th}$ figures), the gradient variance curves suffer from heavy misalignment between modules. Specifically, the gradient variances are significantly small in the RPN, FPN, detection head, and mask head. We find that the larger the variance gap, the lower the model performance (the best performance is achieved when batch size equals 32). **Second row**: In figures from left to right, we compare the performance (right vertical axis) and training time of AGVM (bar diagram, left vertical axis) in different visual tasks, including object detection ($1^{st}$ figure), instance segmentation ($2^{nd}$), panoptic segmentation ($3^{rd}$), and semantic segmentation ($4^{th}$), where the models are trained using different methods with different batch sizes. The "×" indicates training failure when using previous methods. Our method outperforms the recent approaches in all tasks with various batch sizes, significantly reducing training time.

# 1 Introduction

The recent successes in many tasks of dense visual predictions rely on the large-scale datasets [1, 2, 3], the increase of computational power (*e.g.,* GPUs), and the parallel training paradigm with large sample batches. Sufficient computational resource enables large-batch training, greatly reducing the training time [4]. However, although simply scaling the batch size allows fewer iterations to update the parameters of deep neural networks, it often leads to dramatic drop of generalization performance [5, 6, 7].

To reduce the generalization gap in the large-batch training paradigm, LARS [8] scales the batch size of a plain ResNet50 from 8k to 32k without losing accuracy, enabling to train an image classification model on ImageNet in a few minutes. However, different from the plain network architectures in ImageNet classification [9, 10, 11], many tasks of dense visual predictions, such as object detection [12, 13, 14, 15, 16] and segmentation [17, 18, 19, 20], are solved by more complicated pipelines, which consist of multiple different modules, such as region proposal network (RPN) [12], feature pyramid network (FPN) [21], detection head, and segmentation head. Nevertheless, the recent advanced large-batch optimization methods such as LARS [8] and LAMB [6] are typically not sufficient to achieve good generalization performance in dense visual predictions. The long training time of dense predictors greatly limits the researchers from making full use of the increasing computational power and large-scale datasets.

To address the above challenge, we present a novel large-batch training algorithm, named Adaptive Gradient Variance Modulator (AGVM), which can train different complicated dense predictors with very large batch size, significantly reducing their training time while maintaining the generalization performance. The design of AGVM is motivated by a training phenomenon overlooked in prior arts. We call it gradient variance misalignment, which would present when a visual dense prediction pipeline contains many different modules and is trained with a large mini-batch, where different modules (*e.g.,* backbone, RPN, FPN, and heads) can have different gradient variance magnitudes, impeding the generalization ability.

As shown in the first row of Fig.1, where Mask R-CNN [17] with ResNet50 [22] as the backbone is trained using different batch sizes, we compare the gradient variances of different network modules, including backbone, FPN, RPN, detection head, and mask head. We see that when the batch size is small (32 in the first figure), the gradient variances of different network modules are similar throughout the training process. When the batch size increases from 256 to 1024 ($2^{\text{nd}} \sim 4^{\text{th}}$ figures), the gradient variances misalign in different modules whose variance gap enlarges during training. Training fails when batch size equals 1024. More importantly, the gradient variances have significantly smaller values in the RPN, FPN, detection head, and mask head compared to that in the backbone, and their gradient variances change sharply in the late stage of training (two figures in the middle). We find that such misalignment undesirably burdens the large-batch training, leading to severe performance drop and even training failure. More observations on various visual tasks and networks can be found in Appendix A.2.

The above empirical analysis naturally inspires us to design a simple yet effective method AGVM for training dense visual predictors with multiple modules using very large batch size. AGVM directly modulates the misaligned variance of gradient, making it consistent between different network modules throughout training. As shown in the second row of Fig.1, AGVM significantly outperforms the recent approaches of large-batch training in four different visual prediction tasks with various batch sizes from 32 to 2048. For example, AGVM enables us to train an object detector with a huge batch size 1536 (where prior arts may fail), reducing training time by more than 35× compared to the regular training setup.

This work makes three main **contributions**. **Firstly**, we carefully design AGVM, which to our knowledge, is the first large-batch optimization method for various dense prediction networks and tasks. We evaluate AGVM in different architectures (*e.g.,* CNNs and Transformers), solvers (*e.g.,* SGD and AdamW), and tasks (*e.g.,* object detection, instance segmentation, semantic segmentation, and panoptic segmentation). **Secondly**, we provide a convergence analysis of AGVM, which converges to a stable point in a general non-convex optimization setting. We also conduct an empirical analysis that reveals an important insight: the inconsistency of effective batch size between different modules would aggravate the gradient variance misalignment when batch size is large, leading to performance drop and even training failure. We believe this insight may facilitate future research for large-scale training of complicated vision systems. **Thirdly**, extensive experiments are conducted to evaluate AGVM, which achieves many new state-of-the-art performances on large-batch training. For example, AGVM demonstrates more stable generalization performance than prior arts under extremely large batch size (*i.e.,* 10k). In particular, it enables training of the widely-used Faster R-CNN+ResNet50 within 4 minutes without performance drop. More importantly, AGVM can train a detector with one billion parameters within just 3.5 hours, which reduces the training time by 20.9×, while achieving a top-ranking mAP 62.2 on the COCO dataset.

## 2 Preliminary and Notation

Let $S = \{(x_i, y_i)\}_{i=1}^{n}$ denote a dataset with $n$ training samples, where $x_i$ and $y_i$ represent a data point and its label respectively. We can estimate the value of a loss function $L : \mathbb{R}^d \to \mathbb{R}$ using a mini-batch of samples that are randomly sampled, and obtain $l(w_t) = \frac{1}{b} \sum_{j \in S_t} L(w_t, (x_j, y_j))$, where $S_t$ denotes the mini-batch at the $t$-th iteration with batch size $|S_t| = b$ and $w_t$ represents the parameters of a deep neural network. We can apply stochastic gradient descent (SGD), one of the most representative algorithms, to update the parameters $w_t$. The SGD update equation with learning rate $\eta_t$ is:

$$w_{t+1} = w_t - \eta_t \nabla l(w_t), \tag{1}$$

where $\nabla l(w_t)$ represents the gradient of the loss function with respect to $w_t$.

**Layerwise Scaling Ratio.** In large-batch training, You et al. [8] observe that the ratio between the norm of the layer weights and the norm of the gradients is unstable (*i.e.,* oscillate a lot), leading to training failure. You et al. [8] present the LARS algorithm, which adopts a layerwise scaling ratio, $\|w_t^{(i)}\| / \|\nabla l(w_t^{(i)}) + \lambda w_t^{(i)}\|$, to modify the magnitude of the gradient of the $i$-th layer $\nabla l(w_t^{(i)})$, where $w_t^{(i)}$ and $\lambda$ indicate the parameters of the $i$-th layer and the weight decay coefficient, respectively. Furthermore, LAMB [6] improves LARS by combining the AdamW optimizer with the layerwise scaling ratio. It can be formulated as $r_t = m_t / \sqrt{v_t + \epsilon}$, where $m_t = \beta_1 m_{t-1} + (1 - \beta_1) \nabla l(w_t)$

and $v_t = \beta_2 v_{t-1} + (1 - \beta_2)\nabla l(w_t)^2$. The layerwise scaling ratio of LAMB can be computed by $\|w_t^{(i)}\|/\|r_t^{(i)} + \lambda w_t^{(i)}\|$.

**Sharpness-aware Minimization.** Large-batch training often converges to a sharp local minima, resulting in undesired generalization performance. The sharpness-aware minimization (SAM) [23] algorithm explicitly penalizes the sharp minima and finds the parameters whose neighbors (in an $l_p$-ball) have low training loss function values using the following objective function:

$$l^{\text{SAM}}(w_t) = \max_{\|\epsilon\|_p \le \rho} l(w_t + \epsilon). \tag{2}$$

To solve the above equation, SAM applies one-step gradient ascent to determine $\epsilon = \rho\nabla l(w_t)/\|\nabla l(w_t)\|$. Its gradient is then approximated by $\nabla l^{SAM}(w_t) \approx \nabla l(w_t)|_{w_t+\epsilon}$. However, SAM involves two sequential gradient computations at each iteration and thus doubles the computational cost.

**Gradient Variance Estimation.** Qin et al. [24] utilize the cosine similarity between two aggregated gradients from the replicas in a distributed training system, to estimate the gradient variance between SGD and GD efficiently. Specifically, we can compute the gradient for each sample in the $t$-th mini-batch $S_t$ of batch size $b$, denoted by $r_{1,t}, ..., r_{j,t}, ..., r_{b,t}$. We have $\nabla l(w_t) = \frac{1}{b}\sum_{j=1}^{b} r_{j,t}$. We split the above gradients into two groups and average each group, obtaining $G_{t,1} = \frac{2}{b}\sum_{j=1}^{\frac{b}{2}} r_{2j-1,t}$ and $G_{t,2} = \frac{2}{b}\sum_{j=1}^{\frac{b}{2}} r_{2j,t}$, respectively. Then the gradient variance can be measured by $\Phi_t = 1 - \cos(G_{t,1}, G_{t,2})$, where $\cos(\cdot, \cdot)$ is the cosine similarity function.

## 3 Our Approach

Our goal is to perform large-batch training for dense visual predictors with many different network modules. As illustrated in Fig.1, the inconsistency of gradient variances among different modules need to be modulated.

**Gradient Variance across Modules.** We derive an updated (considering learning rate) gradient variance to delve into the difference of network modules in complicated dense visual prediction pipelines. The updated gradient variance of the $i$-th network module at the $t$-th iteration can be formulated as:

$$\text{Var}(\eta_t g_t^{(i)}) = \frac{n-b}{2n-b} \eta_t^2 \underbrace{(1 - \mathbb{E}[\cos(G_{t,1}^{(i)}, G_{t,2}^{(i)})])}_{\Phi_t^{(i)}} \mathbb{E}[\|g_t^{(i)}\|^2], \tag{3}$$

where $n$ and $b$ are the number of training samples and the mini-batch size, respectively. $\eta_t$ is the learning rate. $g_t^{(i)}$ indicates the gradient of the $i$-th network module. $G_{t,1}^{(i)}$ and $G_{t,2}^{(i)}$ are two groups of the gradient estimation as discussed above. Since each entry in the vector $g_t^{(i)}$ could be assumed *i.i.d.* in a massive dataset following [24, 25], $\Phi_t^{(i)}$ is thus proportional to the above updated gradient variance. At each training iteration, we can approximate the updated gradient variance by $\Phi_t^{(i)} = \eta_t^2(1 - \cos(G_{t,1}^{(i)}, G_{t,2}^{(i)}))$. Note that $\Phi_t^{(i)}$ for $i$-th module has been normalized by the number of parameters, so $\Phi_t^{(i)}$ of different modules are comparable. For consistency of presentation, we still call $\Phi_t^{(i)}$ gradient variance, which enables us to estimate the gradient variance of each network module at each training iteration. More discussions can be found in Appendix A.1.

**Adaptive Gradient Variance Modulator (AGVM).** Let $\mathcal{M}$ be a set of modules in a complicated dense prediction pipeline, where $\mathcal{M}$ has $h$ different modules. At the $t$-th iteration, we have a set of learning rates, $\{\hat{\eta}_t^{(i)}|i \in \{1, 2, ..., h\}\}$, corresponding to different modules. We treat the `Backbone` ($i = 1$) as the anchor and modulate other modules making their gradient variances consistent with the `Backbone`. Specifically, we adjust the module learning rates $\hat{\eta}_t^{(i)}$ by using the ratio between $\Phi_t^{(1)}$ and $\Phi_t^{(i)}$. The update rule for each network module can be written as:

$$w_{t+1}^{(i)} = w_t^{(i)} - \hat{\eta}_t^{(i)} g_t^{(i)}, \text{ where } \hat{\eta}_t^{(i)} = \eta_t \mu_t^{(i)} \text{ and } \mu_t^{(i)} = \sqrt{\frac{\Phi_t^{(1)}}{\Phi_t^{(i)}}}, \tag{4}$$

Table 1: **Comparisons between different methods.** "Generalization" indicates the methods' generalization ability for dense visual prediction tasks. The number of "+" in the column "stable to batch size scaling" means the degree of stability when batch size is increased, whereas the number in the bracket means the maximum applicable batch size without divergence on object detection. We measure the average extra overhead of the Faster R-CNN+ResNet50 detector at each iteration using 128 NVIDIA A100 GPUs (total batch size is 1024). The number in the column "extra overhead" indicates the ratio of extra overhead (an extra all-reduce call) compared to the original computations. "N/A" means no extra overhead.

| Method | Solution | Generalization | Less hyperparam. tuning | Stable to batch size scaling | Extra overhead |
|---|---|---|---|---|---|
| MegDet [28] | Accumulate statistics of BN | ✔ | ✔ | + (1024) | N/A |
| SAM [23] | Penalize sharp minima | ✗ | ✗ | + (2048) | 100% |
| LARS [8] | Rectify layerwise gradient | ✗ | ✗ | + (1024) | N/A |
| LAMB [6] | Rectify layerwise gradient | ✗ | ✗ | ++ (4096) | N/A |
| PMD-LAMB [29] | Reduce historical effect | ✔ | ✗ | ++ (4096) | N/A |
| AGVM | Balance gradient variance | ✔ | ✔ | +++ (10k) | 0.12% |

where $\eta_t$ is the global learning rate. However, simply adjusting the learning rates on-the-fly would easily yield training failure due to the transitory large variance ratio that impedes the optimization. We propose a momentum update to address this problem. Let $\alpha \in [0, 1)$ be a momentum coefficient, we have:

$$\mu_t^{(i)} \leftarrow \alpha \mu_{t-1}^{(i)} + (1 - \alpha)\mu_t^{(i)}, \tag{5}$$

which can reduce the influence of unstable variance. Note that we update $\mu_t^{(i)}$ each $\tau$ iterations.

**Discussion on Momentum and Weight Decay.** In practice, the weight decay is widely used as a regularizer and is tightly coupled with the learning rate and the momentum. For instance, the gradient $g_t^{(i)}$ will be replaced by the momentum, such as $m_t^{(i)} = \beta_1 m_{t-1}^{(i)} + (1 - \beta_1)(g_t^{(i)} + \lambda w_t^{(i)})$ [6, 26], where $\beta_1$ and $\lambda$ indicate the momentum coefficient and the weight decay coefficient, respectively. We observe that it's also important to modulate the learning rate by Eq.(4) when weight decay is presented. In addition, since the above $m_t$ is a momentum-based moving average of $(g_t^{(i)} + \lambda w_t^{(i)})$, we can directly apply $\hat{\eta}_t^{(i)}$ onto $m_t^{(i)}$.

**Extensions to Different Optimization Algorithms.** AGVM can be easily embedded into different optimization algorithms such as SGD and AdamW. We demonstrate the details in Appendix A.6: Alg.1 and Alg.2, respectively. They can be easily implemented using a deep learning framework *e.g.,* PyTorch [27].

**Discussion on Convergence Rate.** With AGVM, the SGD and the AdamW optimizers still have appealing convergence properties in the general non-convex settings. Considering some mild assumptions in stochastic optimization and the case without heavy-ball momentum ($\beta_1 = 0$), SGD and AdamW achieve $O(1/\sqrt{T})$ and $O(\ln(T)/\sqrt{T})$ convergence rate respectively with appropriate choice of the learning rate $\eta_t$. We present the analysis in Appendix A.4.

**Comparisons with Existing Works.** The purpose of exploring large-batch training is to speed up model training with increasing computational power, as well as enabling us to explore the larger dataset. As shown in Table 1, the seminal works such as LARS [8], LAMB [6], and SAM [23] have made great contributions to large-batch training for plain vision pipelines *e.g.,* image-level prediction, despite that they often require hyper-parameter tuning by experienced engineers. For complicated pipelines of dense visual predictions, they are typically not sufficient to achieve desired generalization performance. MegDet [28] and PMD-LAMB [29] contribute the preliminary attempts by applying large-batch training on object detection. Different from these approaches, we revisit the design paradigm of the complicated dense visual perception pipelines and present a simple yet effective solution, AGVM, which is insensitive to hyperparameter tuning and can be easily plugged into many visual perception pipelines. For example, AGVM can perform stable training with an unprecedented batch size 10K, which could greatly reduce the training time. Moreover, AGVM adds a negligible computational overhead in training, unlike SAM which involves two sequential (non-parallelizable) gradient computations at each iteration, resulting in a significant increase of the training time.

Table 2: **Comparisons** in different tasks (*i.e.,* object detection, instance segmentation, semantic segmentation, and panoptic segmentation) and pipelines (*i.e.,* Faster R-CNN, Mask R-CNN, Semantic FPN, and Panoptic FPN). All pipelines use ResNet50 as the backbone and we use SGD as optimizer. We see that previous methods' performances drop a lot when scaling the batch size and even result in training failure when batch size is 1024 ("NaN"). Since LARS always leads to huge performance drop in large-batch settings, so we only report its performance on Mask R-CNN. We also report the comparisons with MegDet and SAM. The best-performing models are shown in bold. Surprisingly, AGVM can alleviate the training difficulties in large-batch settings.

| Pipeline | Dataset | Task | Batch size | Performance | | | | Iterations |
| --- | --- | --- | --- | --- | --- | --- | --- | --- |
| | | | | MegDet | SAM | LARS | Ours | |
| Faster R-CNN | COCO | Detection | 32 | 36.8 | 36.0 | - | **36.8** | 58640 |
| | | | 256 | 36.1 | 36.5 | - | **36.7** | 7344 |
| | | | 512 | 35.8 | 35.7 | - | **36.7** | 3680 |
| | | | 1024 | 34.2 | 33.0 | - | **35.4** | 1840 |
| Mask R-CNN | COCO | Instance Seg | 32 | 33.9 | 33.7 | **34.0** | 33.9 | 51310 |
| | | | 256 | 33.7 | 33.9 | 32.0 | **34.1** | 6426 |
| | | | 512 | 33.1 | 33.0 | 30.4 | **33.9** | 3220 |
| | | | 1024 | NaN | 31.0 | 25.1 | **32.6** | 1610 |
| Semantic FPN | ADE20K | Semantic Seg | 32 | 37.5 | **38.8** | - | 37.5 | 160000 |
| | | | 512 | 36.7 | **37.6** | - | 37.3 | 10000 |
| | | | 1024 | 36.4 | **37.5** | - | 37.3 | 5000 |
| | | | 2048 | 36.2 | 36.2 | - | **37.0** | 2500 |
| Panoptic FPN | COCO | Panoptic Seg | 32 | 38.9 | **39.0** | - | 38.9 | 51310 |
| | | | 256 | 39.2 | 39.3 | - | **39.3** | 6426 |
| | | | 512 | 38.7 | 38.7 | - | **39.5** | 3220 |
| | | | 1024 | NaN | NaN | - | **38.8** | 1610 |

# 4 Experiments

**Dataset.** We conduct comprehensive experiments on the MS-COCO 2017 [2] and the ADE20K [30] datasets. Specifically, we perform various tasks of object detection, instance segmentation, and panoptic segmentation on COCO, and conduct semantic segmentation on ADE20K.

**Baselines.** Since the prior arts of large-batch optimization methods can be divided into two types, SGD-based methods (*i.e.,* LARS [8], MegDet [28]) and AdamW-based methods (*i.e.,* LAMB [6], PMD-LAMB [29]). For fair comparison, we introduce two training configurations using SGD and AdamW with AGVM, respectively. The details of the hyper-parameter settings can be found in Appendix A.5.

**Pipelines and Models.** To evaluate the generalization ability of AGVM, we conduct extensive experiments on different pipelines, including RetinaNet [31], Faster R-CNN [12], Mask R-CNN [17], Panoptic FPN [32], and Semantic FPN [32]. For the backbone networks, we use ResNet [22] and Swin Transformer [33]. We strictly follow the official implementations of these pipelines and models.

**Implementation Details.** We implement AGVM in PyTorch and reproduce PMD-LAMB with the official implementation of LAMB [6]. We also evaluate LARS [8] and SAM [23] by borrowing their official implementations. To make fair comparisons, we follow the same learning rate scaling method in all experiments. For SGD optimizer, we use linear learning rate scaling when batch size is less than 128 (256 on semantic segmentation). When the batch size is greater than 128, we use the square root of learning rate scaling to avoid divergence in the training process. For PMD-LAMB and LAMB, we follow the learning rate scaling scheme in [29]. We apply a learning rate warm-up scheme to avoid divergence when the learning rate is large. The implementation details can be found in Appendix A.5.

## 4.1 Comparisons to the State-of-The-Art Methods

Table 3 compares the results of object detection on the COCO dataset with different backbones and batch sizes. We compare the mAP and the number of iterations of LAMB, PMD-LAMB, and AGVM using the AdamW optimizer. To our knowledge, AGVM reports the first result that successfully scales the batch size to 1536 with negligible performance drop compared to small-batch training using LAMB. We also see that AGVM contributes significant improvements along with the continuous increase of the batch size. By scaling the batch size larger than 1024 for different backbones, AGVM

Table 3: **Comparisons** of performance for object detection on the COCO dataset with different backbones and batch sizes. We compare the mAP and the number of iterations of AdamW, LAMB, PMD-LAMB, and AGVM+AdamW. The best-performing models are shown in bold. The underlined numbers indicate the results are borrowed directly from [29].

| Pipeline | Backbone | Batch size | Performance | | | | Iterations |
|---|---|---|---|---|---|---|---|
| | | | AdamW | LAMB | PMD-LAMB | AGVM (ours) | |
| Faster R-CNN | ResNet50 | 32 | 37.1 | 36.7 | 36.7 | **37.1** | 43980 |
| | | 256 | 36.9 | 36.2 | 36.7 | **37.2** | 5508 |
| | | 512 | 36.2 | 35.5 | 36.5 | **36.8** | 2760 |
| | | 1024 | 36.2 | 34.8 | 35.3 | **37.0** | 1380 |
| | | 1536 | 35.9 | 33.2 | 33.5 | **36.6** | 924 |
| Faster R-CNN | Swin-Tiny | 32 | 43.6 | 42.9 | 40.2 | **43.7** | 47645 |
| | | 256 | 43.4 | 43.5 | 42.4 | **43.5** | 5967 |
| | | 512 | 42.7 | 42.9 | 41.3 | **43.2** | 2990 |
| | | 1024 | 42.4 | 41.6 | 39.4 | **42.8** | 1495 |

can still achieve 36.6 and 42.8 mAP without heavy hyper-parameter tuning. In conclusion, compared with LAMB and PMD-LAMB, AGVM achieves more accurate results whilst reducing training iterations and runtime. AGVM can be embedded in CNN and Transformer models.

**Generalize to various pipelines, architectures, and optimizers.** AGVM can be generalized to different tasks, pipelines, architectures, and optimizers. Table 2 compares MegDet, SAM, LARS, and AGVM in different dense visual prediction tasks, including object detection, instance segmentation, semantic segmentation, and panoptic segmentation on COCO and ADE20K. We evaluate four representative pipelines (*e.g.,* Faster R-CNN, Mask R-CNN, Semantic FPN, and Panoptic FPN) with different batch sizes from 32 to 1024. We see that scaling the batch size only allows fewer iterations to update weights in previous methods, whose performances drop a lot and even have training failure when the batch size is 1024 (denoted by "NaN"). In contrast, AGVM yields surprising results in all tasks when increasing the batch size. Table 6 reports the performances of AGVM trained with different optimizers, SGD and AdamW. AGVM works well with both of them.

Table 4: Training time of Faster R-CNN with batch size 2 per NVIDIA A100.

| Batch size | 32 | 256 | 512 | 1024 | 1536 |
|---|---|---|---|---|---|
| GPUs | 16 | 128 | 256 | 512 | 768 |
| Time (min) | 148 | 20.8 | 11.8 | 6.0 | **4.2** |

Table 5: Scaling the batch size to 10k on RetinaNet with ResNet18.

| Batch size | 32 | 4k | 10k |
|---|---|---|---|
| PMD-LAMB | 31.4 | 23.5 | NaN |
| Ours | 32.8 | **28.7** | **26.7** |

Table 6: AGVM+different optimizers on Faster R-CNN. AGVM works well with both these optimizers.

| Optimizer | AGVM | Batch size | Backbone | mAP |
|---|---|---|---|---|
| SGD | ✗ | 512 | ResNet50 | 35.8 |
| SGD | ✔ | 512 | ResNet50 | **36.7** |
| AdamW | ✗ | 512 | ResNet50 | 36.2 |
| AdamW | ✔ | 512 | ResNet50 | **36.8** |
| AdamW | ✗ | 512 | Swin-Tiny | 42.7 |
| AdamW | ✔ | 512 | Swin-Tiny | **43.2** |

Table 7: Anchor module selection. We report the segmentation mAP with different anchor modules.

| Pipeline | Modules | mAP |
|---|---|---|
| Mask R-CNN | Backbone | **33.9** |
| | FPN | 33.3 |
| | Detection Head | 33.1 |
| | RPN | 33.1 |
| | Mask Head | 32.9 |

**Training COCO in 4.2 minutes.** With AGVM, we can push the frontier of fast training time on COCO. We employ Faster R-CNN with ResNet50-FPN as the detector and use the same experimental setting as [29]. Then we explore how fast AGVM can reach the 36.6 mAP@0.5:0.95 reported in [29] (which needs 12 minutes to train). Different from the hardware setup in Fig. 1 (batch size 8 per GPU), this experiment is conducted on 768 NVIDIA A100 GPUs. As shown in Table 4, we reduce the original small-batch training time from 2.5 hours to only 4.2 minutes, which is the fastest record to our knowledge.

**Scaling the batch size to 10k.** We also try to push the frontier of large batch size in dense visual prediction tasks. We choose RetinaNet with ResNet18 as the detector, which is trained for 24 epochs

Table 8: **Extending UniNet [34] to one billion parameters**. Both AdamW and PMD-LAMB do not converge when the batch size is 960. On the contrary, our method achieves a top-ranking mAP 62.2 on the COCO dataset, while reducing the training time by 20.9×.

| Optimizer | Batch size | Box mAP | Seg mAP | Iterations | Wall-clock time |
|-----------|-----------|---------|---------|-----------|-----------------|
| AdamW | 32 | 62.6 | 53.8 | 43980 | 73 hours |
| AdamW | 960 | NaN | NaN | - | - |
| PMD-LAMB | 960 | NaN | NaN | - | - |
| Ours | 960 | 62.2 | 53.4 | **1349** | **3.5 hours** |

Table 9: **Insensitive to hyper-parameter $\tau$ and $\alpha$.** We gradually decrease the update frequency of $\mu_t^{(i)}$ from left to right and report the Detection mAP and Segmentation mAP of Mask R-CNN. These results indicate AGVM is not sensitive to these two hyper-parameters. However, when we don't introduce moving average coefficient, the training fails in the early stage.

| $\tau$ / $\alpha$ | None | 5 / 0.95 | 5 / 0.97 | 10 / 0.97 | 20 / 0.97 | 20 / 0.98 |
|-----------|------|----------|----------|-----------|-----------|-----------|
| mAP | NaN | 37.5 / 33.9 | 37.5 / 34.0 | 37.5 / 33.9 | 37.6 / 33.9 | 37.5 / 34.0 |

(2×) using the AdamW optimizer. For batch size 4k and 10k, the learning rates are 0.001 and 0.0015, respectively. The mAP results on COCO are shown in Table 5. Without bells and whistles, the batch size is successfully scaled to 10k while maintaining generalization ability, but PMD-LAMB fails ("NaN").

**Scaling the detector to 1-Billion parameters.** We evaluate AGVM on an extremely-large detector using the UniNet [34]. We extend it to one billion parameters by following the design in [34]. The detailed settings are released in Appendix A.5. Table 8 shows that AGVM still stabilizes and accelerates the training process in such a large model regime. Both AdamW and PMD-LAMB diverge in the early training stage. AGVM can reduce the training time from 3 days (batch size 32) to 3.5 hours using 480 NVIDIA A100 GPUs, achieving a 62.2 box mAP on COCO test-dev benchmark, whilst reducing the training wall-clock time by more than 20 times.

## 4.2 Ablation Study

**Insensitive to hyper-parameter $\tau$ and $\alpha$.** We study the effect of the interval parameter $\tau$, which means we update $\mu_t^{(i)}$ every $\tau$ iterations, as well as the coefficient of moving average $\alpha$ using Mask R-CNN. The experimental results in Table 9 indicate that AGVM is not sensitive to these two hyper-parameters. In practice, we employ $\tau = 10$ and $\alpha = 0.97$ by default. When the batch size is significantly large (*e.g.,* larger than 1K), we reduce the interval to $\tau = 5$ to update $\mu_t^{(i)}$ faster.

**Anchor module selection.** In AGVM, we choose the backbone network as the anchor and modulate other modules to make their gradient variances consistent with the backbone. To deeply investigate this selection, we choose different modules as the anchors. As shown in Table 7, we see that the backbone is the optimal anchor because the backbone network plays the most important role in dense visual predictions.

**Delving into the gradient variance misalignment.** We answer an important question: *what causes the gradient variance misalignment for dense visual predictors?* To tackle this question, we revisit the data flow of dense prediction pipelines and find that the effective batch size is not consistent between different network modules. For instance, due to the shared detection head (*i.e.,* classifiers and regressors) in all the levels of the FPN and different region proposals, the detection head has a different effective batch size compared to the backbone. Similarly, the RPN (or detection head in RetinaNet) shared by all FPN levels and pixel-wise loss computation lead to the increased effective batch size in RPN. Similar to a previous work [25], we find that a larger effective batch size leads to lower gradient variance of modules (*e.g.,* RPN, detection head).

To explore these analyses, we conduct a progressive ablation study using the RetinaNet, as shown by the different gradient variance curves in Fig.2. We have three observations. (1) Intuitively, the shared head leads to the unavoidable batch size misalignment between the backbone and the detection head. For example, given an input mini-batch size $B$, the valid mini-batch size for the detection head is $NB$, where $N$ is the pyramidal feature number. This motivates us to directly replace the shared detection head by independent detection heads. As illustrated by the second figure in Fig.2, the gradient variance misalignment between the detection head and the backbone has been significantly reduced. (2) Furthermore, compared with the plain network architecture, we argue that the effective

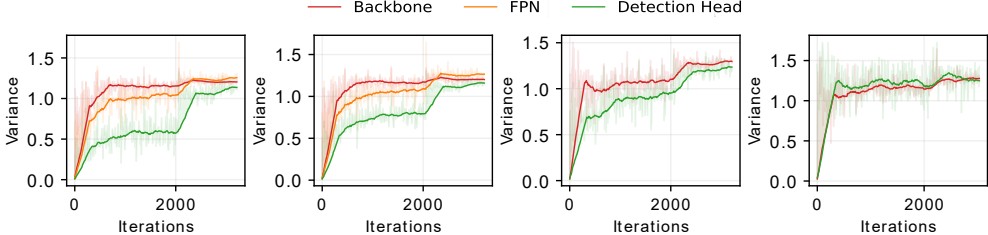

Figure 2: **Ablative experiments on exploring the gradient variance misalignment.** To validate our result on effective batch size, we progressively use independent detection heads, remove FPN, and mask 75% pixels to reduce the effective batch size on the detection head. Finally, we find a near-constant trend of variance throughout training towards convergence between the backbone and the detection head.

batch size is also related to the bottom-up and top-down pathways in FPN. To evaluate this, we remove FPN and only adopt the final-level feature map to perform detection. As shown in the third figure in Fig.2, this alleviates the variance difference between the backbone and the detection head. (3) In the fourth figure, we randomly ignore 75% pixels for loss computation in the predictions generated by detection head. This leads to a near-constant trend of variance throughout training towards convergence between the backbone and the detection head. We have done a similar study using Faster R-CNN, whose results and discussions can be found in Appendix A.3.

## 5 Related Work

We review related work on two aspects including dense visual predictions and large-batch optimization. We defer the related work on dense visual predictions to Appendix A.7.

**Large-batch Optimization.** For large scale deep model training, it is significant to adopt a larger batch size for better hardware utilization and system scalability. However, large-batch training is prone to converge to a sharp minima, resulting in undesired generalization ability [7]. The main reason is that the number of iterations will decrease when we fix the number of epochs in large-batch settings. Researchers [35, 36] try to carefully tune the hyper-parameters to narrow this generalization gap. In detail, by incorporating learning rate warm-up and linear scaling, Goyal et al. [5] successfully train ResNet50 with batch size 8192 without loss in generalization performance. Recently, to avoid these hand-tuned methods, the adaptive learning rate on large-batch training has gained enormous attention from researchers. For example, LARS and LAMB algorithms [8, 6] enable researchers to scale the batch size for ResNet50/BERT to 32k/64k. Both LARS and LAMB leverage the norm of weights and gradients to adjust the learning rate of each layer. These adaptive methods enable researchers to train ImageNet in a few minutes [37, 38, 39]. Johnson et al. [40] propose AdaScale SGD, a novel learning rate schedule rule for stabilizing the warm-up stage. However, it highly depends on the parallelism degree of the system. Liu et al. [41] use adversarial learning to further scale the batch size to 96k. More recently, sharpness-aware minimization (SAM) [23] introduces a procedure to minimize the loss value and loss sharpness to close the generalization gap. However, SAM suffers from training efficiency since the update rule of SAM involves two sequential gradient computation at each iteration. There are few works [42, 43] towards improving the efficiency of SAM. Recently, effort [44] has been made on how to choose an appropriate batch size and corresponding learning rate for large-batch training. And Qin et al. [24] propose Simigrad, which utilizes a lightweight and automated adaptive batching method to enable fine-grained adaptive batch size. However, rather than classification tasks, there are few works towards large-batch training for object detection. Peng et al. [28] implement cross-GPU batch normalization to stabilize the training process and Wang et al. [29] propose PMD-LAMB to reduce the negative effects of the lagging historical gradients. They can scale the training of widely used Faster R-CNN+ResNet50 Detector with batch size 256/1056 with small performance drop.

# 6   Conclusion

The complicated pipelines of dense visual predictions suffer from heavy performance drop in large-batch training. In this paper, we propose and fully study AGVM, which enables module-wise learning rate scaling and successfully scales the batch size to larger than 10K with desired generalization performance. We also provide a convergence analysis, showing that AGVM+SGD and AGVM+AdamW both converge to a stable point in the general non-convex setting. Furthermore, we have conducted extensive experiments to show that AGVM can generalize to different complicated pipelines and challenging tasks, including object detection, instance segmentation, semantic segmentation, and panoptic segmentation. We report unprecedented better performance on large-batch training with very large batch size. For example, AGVM trains Faster R-CNN+ResNet50 using batch size of 1536 in 4.2 minutes without loss of performance. By increasing the object detector UniNet to one billion parameters, AGVM can achieve 62.2 mAP on COCO using a batch size of 960 in just 3.5 hours, reducing the training time by $20.9\times$ compared to the normal small-batch training.

**Limitation and Potential Negative Societal Impact.** Module partitioning is important to estimate the effective batch size quantitatively. For some pipelines without explicit modularity such as the heatmap-based pose estimation, we need to do more empirical analysis. We will investigate it in the future. The potential negative social impact is to use the proposed algorithm to speed up the training of fraud models such as DeepFake [45].

## Acknowledgments and Disclosure of Funding

Ping Luo is supported by the General Research Fund of HK No.27208720, No.17212120, and No.17200622.

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
