# Large-batch Optimization for Dense Visual Predictions

**Zeyue Xue**[*]
The University of Hong Kong
xuezeyue@connect.hku.hk

**Jianming Liang**[*]
Beihang University
ljmmm1997@gmail.com

**Guanglu Song**
Sensetime Research
songguanglu@sensetime.com

**Zhuofan Zong**[*]
Beihang Univerisity
zongzhuofan@gmail.com

**Liang Chen**[*]
Peking University
clandzyy@pku.edu.cn

**Yu Liu**[†]
Sensetime Research
liuyuisanai@gmail.com

**Ping Luo**[†]
The University of Hong Kong,
Shanghai AI Laboratory
pluo@cs.hku.hk

## A   Appendix

For presenting the details in appendix, we extend the notations as: given a module set $\mathcal{M}$, e.g., $\mathcal{M} = \{$Backbone, FPN, RPN, Detection head$\}$ for Faster R-CNN, we define $w = \left\{ w^{(i)} \mid i \in [1, h] \right\}$ as the weights of it, where $h$ means the number of modules in $\mathcal{M}$ and $w^{(i)}$ indicates the learnable parameters of $i$-th module. Let $w \in \mathbb{R}^d$, $w^{(i)} \in \mathbb{R}^{d_i}$, and $\Sigma_{i=1}^{h} d_i = d$. Given a dataset $S = \{(x_i, y_i)\}_{i=1}^{n}$ with $n$ training samples, where $x_i$ and $y_i$ denote a data point and its label respectively, we can estimate a loss function $L : \mathbb{R}^d \to \mathbb{R}$ for a randomly sampled mini-batch $S_t$ to obtain $l(w_t) = \frac{1}{b} \sum_{j \in S_t} L(w_t, (x_j, y_j))$, where $S_t$ is the mini-batch samples with batch size $|S_t| = b$ at the $t$-th iteration. At the $t$-th backward propagation step, we can derive the gradient $\nabla_i l(w_t)$ to update $i$-th module in $\mathcal{M}$. Keep this in mind, we further formulate the gradient of full batch (total samples in $S$) as $\nabla f(w_t)$, where $\nabla f(w_t) = \frac{1}{n} \sum_{j \in S} \nabla L(w_t, (x_j, y_j))$. Naturally, we have $\mathbb{E}[\nabla_i l(w_t)] = \nabla_i f(w_t)$. For convenience, we use $g_t, \|\cdot\|$ and $\|\cdot\|_1$ to denote $\nabla l(w_t), l_2$-norm and $l_1$-norm, respectively. In particular, $g_t^{(i)}$ is used to denote $\nabla_i l(w_t)$.

### A.1   Gradient Variance Estimation

We introduce the gradient variance to measure the gap between SGD (stochastic gradient descent with mini-batch) and GD (gradient descent with full batch). However, computing the accurate gradient variance requires extremely high computational cost and it will slow down training speed dramatically. To address this problem, Qin et al. [1] utilize the cosine similarity between two aggregated gradients from the replicas in a distributed training system to estimate the gradient variance between SGD and GD efficiently. Specifically, we can compute the gradient for each sample in the $t$-th mini-batch $S_t$ of batch size $b$, denoted by $r_{1,t}, ..., r_{j,t}, ..., r_{b,t}$, then we have $g_t = \frac{1}{b} \sum_{j=1}^{b} r_{j,t}$. Since we split the above gradients into two groups, averaging each group can obtain $G_{t,1} = \frac{2}{b} \sum_{j=1}^{\frac{b}{2}} r_{2j-1,t}$ and $G_{t,2} = \frac{2}{b} \sum_{j=1}^{\frac{b}{2}} r_{2j,t}$, respectively. It formulates the gradient variance as:

$$\text{Var}(g_t) = \mathbb{E}[\|g_t - \nabla f(w_t)\|^2] = \frac{n-b}{2n-b}(1 - \mathbb{E}[cos(G_{t,1}, G_{t,2})])\mathbb{E}[\|g_t\|^2], \qquad (1)$$

---

[*]Work done during an internship at Sensetime Research.
[†]Corresponding authors.

36th Conference on Neural Information Processing Systems (NeurIPS 2022).

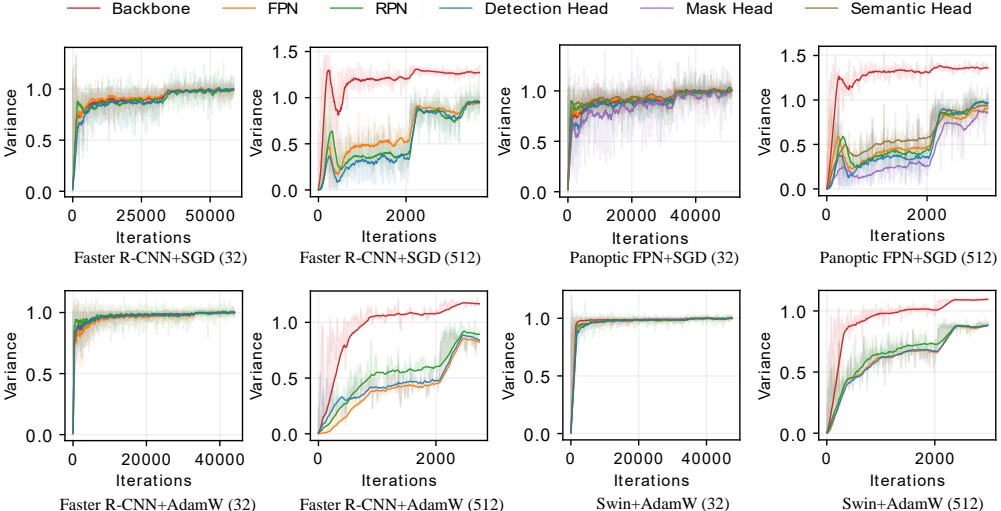

Figure 1: **Comparisons** of the gradient variances (omitting the learning rate $\eta_t$ referring to Eq. (2)) in different modules of different pipelines (*i.e.,* Faster R-CNN and Panoptic FPN) and optimizers (*i.e.,* SGD and AdamW). The number in the bracket represents the batch size. We see that when the batch size is small (*i.e.,* 32), the gradient variances are similar. When the batch size is large (*i.e.,* 512), the gradient variances all suffer significant misalignment of different modules. All pipelines use ResNet50 as the backbone network other than the last two figures, where we adopt Faster R-CNN+Swin-Tiny to visualize the variances.

where $n$ and $b$ are the number of training samples and the mini-batch size, respectively. Then we derive a updated (considering learning rate) gradient variance to delve into the difference of network modules in complicated dense visual prediction pipelines. The updated gradient variance of the $i$-th network module at the $t$-th iteration can be formulated as:

$$\text{Var}(\eta_t g_t^{(i)}) = \mathbb{E}[\|\eta_t g_t^{(i)} - \eta_t \nabla_i f(w_t)\|^2] = \frac{n-b}{2n-b} \underbrace{\eta_t^2 (1 - \mathbb{E}[\cos(G_{t,1}^{(i)}, G_{t,2}^{(i)})])}_{\Phi_t^{(i)}} \mathbb{E}[\|g_t^{(i)}\|^2], \quad (2)$$

where $\eta_t$ is the learning rate. $G_{t,1}^{(i)}$ and $G_{t,2}^{(i)}$ are two groups of the gradient estimation as discussed above for $i$-th submodule. Following [1, 2], since each entry in the vector $g_t^{(i)}$ could be assumed independent and identically distributed (*i.i.d.*) in a massive dataset, $\Phi_t^{(i)}$ is thus proportional to the above updated gradient variance. At each training iteration, we can approximate the updated gradient variance by $\Phi_t^{(i)} = \eta_t^2 (1 - \cos(G_{t,1}^{(i)}, G_{t,2}^{(i)}))$, where $\Phi_t^{(i)}$ indicates the $\text{Var}(\eta_t g_t^{(i)})$ normalized by the number of parameters. For consistency of presentation, we still call $\Phi_t^{(i)}$ gradient variance, which enables us to estimate the gradient variance of each network module at each training iteration. Note that gradient variance magnitude has great influence on the generalization ability of deep neural network [2].

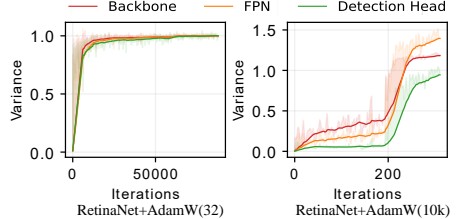

Figure 2: **Comparisons** of variances for RetinaNet with batch size 32 and 10k.

## A.2 Overview of Gradient Variance of Different Pipelines

In this section, we give an overview of the gradient variance comparisons of different pipelines in Fig. 1, including four pipelines (*i.e.,* Faster R-CNN and Panoptic FPN) and two optimizers (*i.e.,* SGD and AdamW). We also show the gradient variances with batch size 32 and 10k in Fig. 2 on RetinaNet. The variances after applying AGVM on Mask R-CNN is shown in Fig. 3.

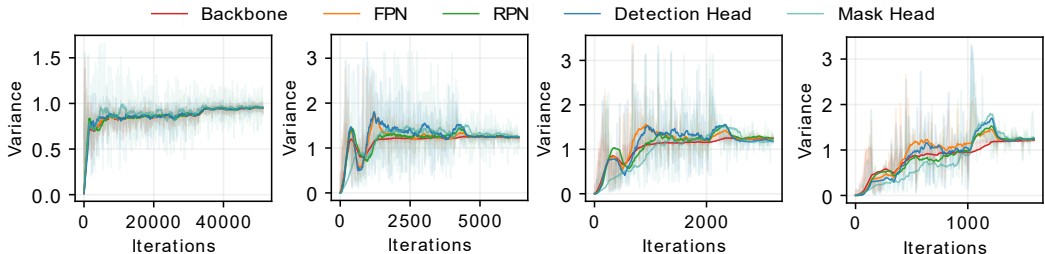

Figure 3: **Comparisons** of the gradient variances of different modules in Mask R-CNN with the help of AGVM. From left to right, the models are trained using SGD with a mini-batch size of 32, 256, 512, and 1024. AGVM helps avoid training failure with batch size 1024.

## A.3 Ablation Study of Variance Misalignment on Faster R-CNN

We define the module set $\mathcal{M}$ as {`Backbone`, `FPN`, `RPN`, `Detection head`} in Faster R-CNN [3] and $|B_i|$ indicates the *effective batch size* of the $i$-th module in $\mathcal{M}$. Intuitively, there are $|B_4| \approx NK|B_1|$ due to the shared detection head (*i.e.,* classifiers/regressors) by all levels of the FPN and different region proposals. $N$ and $K$ indicate the number of FPN levels and region proposals fed into the detection head. To evaluate this assumption, as shown in Fig. 4, we have three observations. (1) Similar to the ablation study on RetinaNet, we remove the FPN and adopt the final level to perform detection. As illustrated by the second figure in Fig. 4, the gradient misalignment phenomenon between detection head and backbone has been reduced. (2) Furthermore, we reduce the number of region proposals from 512 to 10. As shown in the third figure in Fig. 4, this also alleviates the variance difference between detection head and backbone. (3) Finally, we freeze the parameters in the detection head and only train RPN and backbone. Similar to the phenomenon on RetinaNet, this also leads to a variance convergence trend throughout the training between RPN and backbone.

## A.4 Proof of Convergence Rate

In this section, we will show that even using AGVM, SGD and AdamW optimizers still enjoy appealing convergence properties. In order to present our analysis, we first need to make some assumptions.

**Assumptions.** We need to assume function $f(w)$ is $L_i - smooth$ with respect to $w^{(i)}$, *i.e.,* there exists a constant $L_i$ such that:

$$\forall x, y \in \mathbb{R}^d, \|\nabla_i f(x) - \nabla_i f(y)\| \leq L_i \|x^{(i)} - y^{(i)}\|, \tag{3}$$

for all $i \in [1, h]$. We use $L = (L_1, \cdots, L_h)^\top$ to denote the $h$-dimensional vector of Lipschitz constants and use $L_{max}$ to denote $\max_i L_i$. We also assume the following bound on different modules' gradient norm via $\mathbb{E}\left[\|g^{(i)}\|^2\right] \leq K\|\nabla_1 f(w)\|^2$. Furthermore, although it's difficult to quantify the effective batch size of different modules, we argue the ratio of effective batch size between different modules should be bounded, so we can assume $1 \leq \frac{\mathbb{E}[\|\Phi_t^{(1)}\|]}{\mathbb{E}[\|\Phi_t^{(i)}\|]} \leq \alpha_u$ for $i \in [1, h]$ and $t \in [1, T]$. For the sake of simplicity, we give convergence results when $\beta_1 = 0$ and ignore the weight decay coefficient ($\lambda = 0$). However, our analysis should extend to the general case as well. We leave this investigation in future work.

### A.4.1 Convergence of AGVM+SGD

For SGD optimizer, we also assume the following bound on the variance in stochastic gradients $\mathbb{E}\left\|g^{(i)} - \nabla_i f(w)\right\|^2 \leq \sigma_i^2$ for all $w \in \mathbb{R}^d$ and $i \in [1, h]$ with effective batch size $b_i$. For component $i$, we have the following update for SGD optimizer:

$$w_{t+1}^{(i)} = w_t^{(i)} - \eta_t \sqrt{\frac{\mathbb{E}[\|\Phi_t^{(1)}\|]}{\mathbb{E}[\|\Phi_t^{(i)}\|]}} g_t^{(i)}. \tag{4}$$

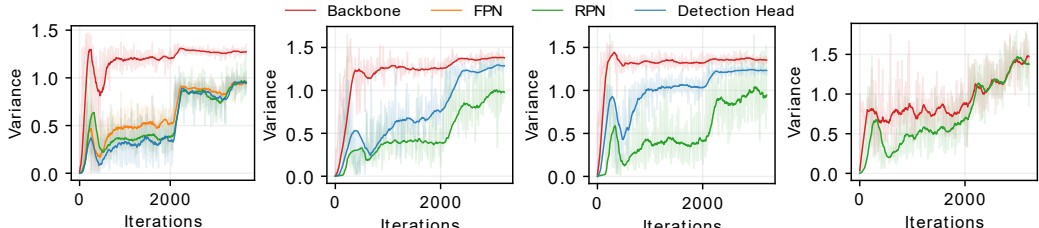

Figure 4: **Ablative experiments on exploring the gradient variance misalignment.** To validate our result on effective batch size, we progressively remove the FPN, decrease region proposals, and freeze the parameters of detection head to reduce the effective batch size. Finally, it also leads to a variance convergence trend throughout the training between RPN and backbone.

Since the function $f$ is $L_i - smooth$, we can obtain the following inequality:

$$f\left(w_{t+1}\right) \leq f\left(w_t\right) + \left\langle \nabla_i f\left(w_t\right), w_{t+1}^{(i)} - w_t^{(i)} \right\rangle + \sum_{i=1}^{h} \eta_t^2 \frac{L_i}{2} \frac{\mathbb{E}[\|\Phi_t^{(1)}\|]}{\mathbb{E}[\|\Phi_t^{(i)}\|]} \left\| g_t^{(i)} \right\|^2. \tag{5}$$

Then, we will first give some analysis on the following ratio:

$$\frac{\mathbb{E}[\|\Phi_t^{(1)}\|]}{\mathbb{E}[\|\Phi_t^{(i)}\|]} = \frac{\mathbb{E}\left[1 - cos(G_{t,1}^{(1)}, G_{t,2}^{(1)})\right]}{\mathbb{E}\left[1 - cos(G_{t,1}^{(i)}, G_{t,2}^{(i)})\right]}. \tag{6}$$

Because the samples are randomly divided into two groups, according to the law of large numbers, when batch size $b$ goes to infinity, we have:

$$\mathbb{E}\left[cos(G_{t,1}^{(j)}, G_{t,2}^{(j)})\right] \to 1, \forall j \geq 1. \tag{7}$$

For $b = 2$, each group only has one sample that comes from the same training distribution, we have:

$$\mathbb{E}\left[cos(G_{t,1}^{(j)}, G_{t,2}^{(j)})\right] \to 0, \forall j \geq 1. \tag{8}$$

Therefore, there exists a $\hat{b}$ that makes the following equation hold,

$$\mathbb{E}\left[cos(G_{t,1}^{(j)}, G_{t,2}^{(j)})\right] \leq \frac{1}{2}, \text{if } b \leq \hat{b}, \forall j \geq 1. \tag{9}$$

Since the effective batch size of backbone is smaller than that of other modules, the gradient variance of backbone is larger than that of other modules, which means:

$$\mathbb{E}\left[cos(G_{t,1}^{(1)}, G_{t,2}^{(1)})\right] \leq \mathbb{E}\left[cos(G_{t,1}^{(i)}, G_{t,2}^{(i)})\right], \forall i > 1. \tag{10}$$

When $b < \hat{b}$, we further have:

$$\mathbb{E}\left[cos(G_{t,1}^{(1)}, G_{t,2}^{(1)})\right]\left(1 - \mathbb{E}\left[cos(G_{t,1}^{(1)}, G_{t,2}^{(1)})\right]\right) \leq \mathbb{E}\left[cos(G_{t,1}^{(i)}, G_{t,2}^{(i)})\right]\left(1 - \mathbb{E}\left[cos(G_{t,1}^{(i)}, G_{t,2}^{(i)})\right]\right), \forall i > 1. \tag{11}$$

Based on this, we have the following:

$$\frac{\mathbb{E}\left[1 - cos(G_{t,1}^{(1)}, G_{t,2}^{(1)})\right]}{\mathbb{E}\left[1 - cos(G_{t,1}^{(i)}, G_{t,2}^{(i)})\right]} \leq \frac{\mathbb{E}\left[cos(G_{t,1}^{(i)}, G_{t,2}^{(i)})\right]}{\mathbb{E}\left[cos(G_{t,1}^{(1)}, G_{t,2}^{(1)})\right]}. \tag{12}$$

By displaying $\delta_t \equiv g_t^{(i)} - \nabla_i f\left(w_t\right)$, we obtain:

$$\mathbb{E}\left[\|g_t^{(i)}\|^2\right] = \mathbb{E}\left[\|\delta_t + \nabla_i f\left(w_t\right)\|^2\right] \leq \sigma_i^2 + \|\nabla_i f\left(w_t\right)\|^2. \tag{13}$$

Following the Eq.(6) in [1], we have:

$$\frac{\|\nabla_i f(w_t)\|^2}{\|\nabla_i f(w_t)\|^2 + \sigma_i^2} \leq \frac{\|\nabla_i f(w_t)\|^2}{\mathbb{E}\left[\|g_t^{(i)}\|^2\right]} = \mathbb{E}\left[cos(G_{t,1}^{(i)}, G_{t,2}^{(i)})\right] \leq 1. \tag{14}$$

With the help of above inequality, we have:

$$\frac{\mathbb{E}\left[cos(G_{t,1}^{(i)}, G_{t,2}^{(i)})\right]}{\mathbb{E}\left[cos(G_{t,1}^{(1)}, G_{t,2}^{(1)})\right]} \le 1 + \frac{\sigma_1^2}{\|\nabla_1 f(w_t)\|^2}. \tag{15}$$

However, as shown in Fig. 2, when the batch size is extremely large (*e.g.,* 10k), we cannot derive the above inequality. In this case, we have:

$$\frac{\mathbb{E}\left[1 - cos(G_{t,1}^{(1)}, G_{t,2}^{(1)})\right]}{\mathbb{E}\left[1 - cos(G_{t,1}^{(i)}, G_{t,2}^{(i)})\right]} \le 1 + \alpha_0 + \frac{\sigma_1^2}{\|\nabla_1 f(w_t)\|^2}, \tag{16}$$

where $\alpha_0$ is a constant that meets $\alpha_u - 1 - \frac{\sigma_1^2}{\|\nabla_1 f(w_t)\|^2} \le \alpha_0 \le \alpha_u - 1$ for all $t \le T$. Then by adding Eq. (16) to Eq. (5), we obtain:

$$f(w_{t+1}) \le f(w_t) + \left\langle \nabla_i f(w_t), w_{t+1}^{(i)} - w_t^{(i)} \right\rangle + \sum_{i=1}^{h} \eta_t^2 \frac{L_i}{2} \left( \alpha_0 + 1 + \frac{\sigma_1^2}{\|\nabla_1 f(w_t)\|^2} \right) \|g_t^{(i)}\|^2. \tag{17}$$

Taking expectation on the both side, according to the assumption on Eq. (6), we have:

$$\mathbb{E}[f(w_{t+1})] \le f(w_t) - \eta_t \sum_{i=1}^{h} \|\nabla_i f(w_t)\|^2 + \sum_{i=1}^{h} \eta_t^2 \frac{L_i}{2} \left( \alpha_0 + 1 + \frac{\sigma_1^2}{\|\nabla_1 f(w_t)\|^2} \right) \mathbb{E}\left[\|g_t^{(i)}\|^2\right]$$

$$\le f(w_t) - \eta_t \sum_{i=1}^{h} \|\nabla_i f(w_t)\|^2 + \sum_{i=1}^{h} \eta_t^2 \frac{L_i}{2} \left( (1 + \alpha_0)\mathbb{E}\left[\|g_t^{(i)}\|^2\right] + K\sigma_1^2 \right)$$

$$\le f(w_t) - \eta_t \sum_{i=1}^{h} \|\nabla_i f(w_t)\|^2 + \sum_{i=1}^{h} \eta_t^2 \frac{L_i}{2} \left( (1 + \alpha_0)(\sigma_i^2 + \|\nabla_i f(w_t)\|^2) + K\sigma_1^2 \right)$$

$$= f(w_t) - \sum_{i=1}^{h} \left( \eta_t - \frac{L_{max}}{2}(1 + \alpha_0)\eta_t^2 \right) \|\nabla_i f(w_t)\|^2 + \sum_{i=1}^{h} \eta_t^2 \frac{L_i}{2} \left( K\sigma_1^2 + (1 + \alpha_0)\sigma_i^2 \right). \tag{18}$$

Summing both sides of this inequality and taking the complete expectation, we get:

$$\mathbb{E}[f(w_{t+1})] \le f(w_1)$$
$$- \sum_{t=1}^{T} \sum_{i=1}^{h} \left( \eta_t - \frac{L_{max}}{2}\eta_t^2(1 + \alpha_0) \right) \mathbb{E}[\|\nabla_i f(w_t)\|^2] + T \sum_{i=1}^{h} \eta_t^2 \frac{L_i}{2} \left( K\sigma_1^2 + (1 + \alpha_0)\sigma_i^2 \right). \tag{19}$$

Define $f_{inf} = \inf f(w_t)$ and arrange the above inequality, we can get:

$$\frac{1}{T} \sum_{t=1}^{T} \sum_{i=1}^{h} \mathbb{E}\left[\|\nabla_i f(w_t)\|^2\right] \le \frac{f(w_1) - f_{inf}}{T\left(\eta_t - \frac{L_{max}}{2}\eta_t^2(1 + \alpha_0)\right)} + \frac{\sum_{i=1}^{h} \eta_t L_i \left( K\sigma_1^2 + (1 + \alpha_0)\sigma_i^2 \right)}{2 - L_{max}\eta_t(1 + \alpha_0)}. \tag{20}$$

Let $\eta_t \le \frac{1}{(1+\alpha_0)L_{max}}$, we have the following bound:

$$\frac{1}{T} \sum_{t=1}^{T} \mathbb{E}\left[\|\nabla f(w_t)\|^2\right] \le \frac{2(f(w_1) - f_{inf})}{T\eta_t} + \sum_{i=1}^{h} \eta_t L_i \left( K\sigma_1^2 + (1 + \alpha_0)\sigma_i^2 \right). \tag{21}$$

### A.4.2 Convergence of AGVM+AdamW

For AdamW optimizer, we also assume $\|g_t\|_\infty \le G - \sqrt{\epsilon}$, $d_i = \frac{d}{h}$. Following [4], we rewrite the learning rate in the following manner: $\tilde{\eta}_t = \eta_t \sqrt{\frac{1-\beta_2^t}{1-\beta_2}}$. Based on this, we can redefine the $v_t$ as

$v_t = \beta_2 v_{t-1} + g_t^2$, and let $\tilde{v}_t = \beta_2 v_{t-1} + \mathbb{E}[g_t^2]$. So the update of original AdamW can be given by:$r_t = \frac{g_t}{\sqrt{v_t + \epsilon}}$, then we have the following update for AGVM+AdamW:

$$w_{t+1}^{(i)} = w_t^{(i)} - \tilde{\eta}_t \sqrt{\frac{\mathbb{E}[\|\Phi_t^{(1)}\|]}{\mathbb{E}[\|\Phi_t^{(i)}\|]}} r_t^{(i)}. \tag{22}$$

Since the function $f$ is $L_i - smooth$, we have the following:

$$f(w_{t+1}) \leq f(w_t) + \left\langle \nabla_i f(w_t), w_{t+1}^{(i)} - w_t^{(i)} \right\rangle + \sum_{i=1}^{h} \tilde{\eta}_t^2 \frac{L_i}{2} \frac{\mathbb{E}[\|\Phi_t^{(1)}\|]}{\mathbb{E}[\|\Phi_t^{(i)}\|]} \left\| r_t^{(i)} \right\|^2. \tag{23}$$

For any component $i$, we have:

$$\mathbb{E}\left[cos(G_{t,1}^{(i)}, G_{t,2}^{(i)})\right] = \frac{\sum_{j=0}^{d_i} (\mathbb{E}\left[g_{t,j}^{(i)}/\sqrt{\epsilon + v_{t,j}^{(i)}}\right])^2}{\sum_{j=0}^{d_i} \mathbb{E}\left[(g_{t,j}^{(i)}/\sqrt{\epsilon + v_{t,j}^{(i)}})^2\right]} \leq 1, \tag{24}$$

where $g_{t,j}^{(i)}$ and $v_{t,j}^{(i)}$ denote the $j$-th entry of $g_t^{(i)}$ and $v_t^{(i)}$. Thanks to the $l_\infty$ bound on $g_t$, we have $g_t^{(i)} \leq \sqrt{\epsilon + v_{t,j}^{(i)}} \leq \frac{G}{\sqrt{1-\beta_2}}$, so that:

$$\frac{\|\nabla_i f(w_t)\|^2}{d_i(G^2/(1-\beta_2))} \leq \frac{\sum_{j=0}^{d_i} (\mathbb{E}\left[g_t^{(i)}/\sqrt{\epsilon + v_{t,j}^{(i)}}\right])^2}{\sum_{j=0}^{d_i} \mathbb{E}\left[(g_t^{(i)}/\sqrt{\epsilon + v_{t,j}^{(i)}})^2\right]} \leq 1. \tag{25}$$

Similar to Eq. (12), then we get:

$$\frac{\mathbb{E}[\|\Phi_t^{(1)}\|]}{\mathbb{E}[\|\Phi_t^{(i)}\|]} = \frac{\mathbb{E}\left[1 - cos(G_{t,1}^{(1)}, G_{t,2}^{(1)})\right]}{\mathbb{E}\left[1 - cos(G_{t,1}^{(i)}, G_{t,2}^{(i)})\right]} \leq \frac{\mathbb{E}\left[cos(G_{t,1}^{(i)}, G_{t,2}^{(i)})\right]}{\mathbb{E}\left[cos(G_{t,1}^{(1)}, G_{t,2}^{(1)})\right]} \leq \frac{d_1(G^2/(1-\beta_2))}{\|\nabla_1 f(w_t)\|^2}. \tag{26}$$

However, since $1 - \beta_2 \to 0$ in general AdamW settings, as well as for some extremely large batch size settings (where the upper bound of Eq. (26) is dominated by $\alpha_u$), we have the following for the sake of consistency:

$$\frac{\mathbb{E}\left[1 - cos(G_{t,1}^{(1)}, G_{t,2}^{(1)})\right]}{\mathbb{E}\left[1 - cos(G_{t,1}^{(i)}, G_{t,2}^{(i)})\right]} \leq \min\{\frac{d_1(G^2/(1-\beta_2))}{\|\nabla_1 f(w_t)\|^2}, \alpha_u\}. \tag{27}$$

We will give the convergence bounds using these two items, respectively. For the first item, by rewriting Lemma 1 in [4], we get:

$$\mathbb{E}\left[\nabla_{i,j} f(w_t) \frac{g_{t,j}^{(i)}}{\sqrt{\epsilon + v_{t,j}^{(i)}}}\right] \geq \frac{(\nabla_{i,j} f(w_t))^2}{2\sqrt{\epsilon + \tilde{v}_{t,j}^{(i)}}} - 2G\mathbb{E}\left[\frac{\left(g_{t,j}^{(i)}\right)^2}{\epsilon + v_{t,j}^{(i)}}\right], \tag{28}$$

where we denote the $j$-th entry of $\nabla_i f(w_t)$ by $\nabla_{i,j} f(w_t)$. Thanks to the $l_\infty$ bounded on $g^{(i)}$, we have:

$$\tilde{\eta}_t \frac{(\nabla_{i,j} f(w_t))^2}{2\sqrt{\epsilon + \tilde{v}_{t,j}^{(i)}}} \geq \frac{\eta_t (\nabla_{i,j} f(w_t))^2}{2G}. \tag{29}$$

Taking expectation on Eq. (23), and adding Eq. (29) to Eq. (23), we have:

$$\mathbb{E}[f(w_{t+1})] \leq f(w_t)$$
$$- \frac{\eta_t}{2G} \|\nabla f(w_t)\|^2 + \sum_{i=1}^{h} \left(2\tilde{\eta}_t G + \frac{\tilde{\eta}_t^2 L_i d_1(G^2/(1-\beta_2))}{2\|\nabla_1 f(w_t)\|^2}\right) \mathbb{E}\left[\left\|r_t^{(i)}\right\|^2\right]. \tag{30}$$

Taking complete expectation on Eq. (30) and sum up:

$$
\begin{aligned}
\mathbb{E}\left[f\left(w_{t+1}\right)\right] \leq &\ f\left(w_1\right) - \frac{\eta_t}{2G} \sum_{t=1}^{T} \mathbb{E}\left[\|\nabla f\left(w_t\right)\|^2\right] \\
&+ \sum_{t=1}^{T} \sum_{i=1}^{h} \left(\frac{2\eta_t G}{\sqrt{1-\beta_2}} \mathbb{E}\left[\left\|r_t^{(i)}\right\|^2\right]\right) + \frac{\eta_t^2 \|L\|_1 d_1 (G^2/(1-\beta_2))KT}{2\epsilon(1-\beta_2)}.
\end{aligned}
\tag{31}
$$

Then, with the help of Lemma 2 in [4], we get:

$$
\begin{aligned}
\mathbb{E}\left[f\left(w_{t+1}\right)\right] \leq &\ f\left(w_1\right) - \frac{\eta_t}{2G} \sum_{t=1}^{T} \mathbb{E}\left[\|\nabla f\left(w_t\right)\|^2\right] \\
&+ \frac{2\eta_t G d}{\sqrt{1-\beta_2}} \left(\frac{1}{T} \ln\left(1 + \frac{G^2}{(1-\beta_2)\epsilon}\right) - T\ln(\beta_2)\right) + \frac{\eta_t^2 \|L\|_1 d_1 (G^2/(1-\beta_2))KT}{2\epsilon(1-\beta_2)}.
\end{aligned}
\tag{32}
$$

For the second item in Eq. (27), taking complete expectation on Eq. (23) and sum up:

$$
\begin{aligned}
\mathbb{E}\left[f\left(w_{t+1}\right)\right] \leq &\ f\left(w_1\right) \\
&- \frac{\eta_t}{2G} \sum_{t=1}^{T} \mathbb{E}\left[\|\nabla f\left(w_t\right)\|^2\right] + \sum_{t=1}^{T} \sum_{i=1}^{h} \left(\left(\frac{2\eta_t G}{\sqrt{1-\beta_2}} + \tilde{\eta}_t^2 \alpha_u \frac{L_i}{2}\right) \mathbb{E}\left[\left\|r_t^{(i)}\right\|^2\right]\right).
\end{aligned}
\tag{33}
$$

With the help of Lemma 2 in [4], we get:

$$
\begin{aligned}
\mathbb{E}\left[f\left(w_{t+1}\right)\right] \leq &\ f\left(w_1\right) - \frac{\eta_t}{2G} \sum_{t=1}^{T} \mathbb{E}\left[\|\nabla f\left(w_t\right)\|^2\right] \\
&+ \left(\frac{2\eta_t G d}{\sqrt{1-\beta_2}} + \tilde{\eta}_t^2 \alpha_u h \frac{\|L\|_1}{2}\right) \left(\frac{1}{T} \ln\left(1 + \frac{G^2}{(1-\beta_2)\epsilon}\right) - T\ln(\beta_2)\right).
\end{aligned}
\tag{34}
$$

Finally, we have:

$$
\begin{aligned}
\frac{1}{2GT} \sum_{t=1}^{T} \mathbb{E}\left[\|\nabla f\left(w_t\right)\|^2\right] \leq &\ \frac{f\left(w_1\right) - f_{inf}}{\eta_t T} + \frac{2Gd}{\sqrt{1-\beta_2}} \left(\frac{1}{T} \ln\left(1 + \frac{G^2}{(1-\beta_2)\epsilon}\right) - \ln(\beta_2)\right) + C, \\
C = \min \Bigg\{ &\ \frac{\eta_t \|L\|_1 dG^2 K}{2\epsilon h(1-\beta_2)^2}, \frac{\eta_t \alpha_u h \|L\|_1}{2(1-\beta_2)} \left(\frac{1}{T} \ln\left(1 + \frac{G^2}{(1-\beta_2)\epsilon}\right) - \ln(\beta_2)\right) \Bigg\}.
\end{aligned}
\tag{35}
$$

For AGVM+SGD, suppose $\eta_t = \frac{1}{\sqrt{T}}$, and for AGVM+AdamW, let $\eta_t = \frac{1}{\sqrt{T}}$ and $\beta_2 = 1 - \frac{1}{T}$, then SGD and AdamW achieve $O(1/\sqrt{T})$ and $O(\ln(T)/\sqrt{T})$ convergence rate, respectively. Note that in this case, the upper bound of Eq. (35) is dominated by the second item of $C$.

### A.4.3 Linear Speedup Property of AGVM

We give the linear speedup property for AGVM+synchronous SGD w.r.t. batch size as a corollary. First, we will prove gradient variance decreases linearly with batch size $b$. For ease of understanding, we assume that $\nabla f(w)$, $g$, $r$ represent the gradient of the full dataset, the mini-batch with size $b$ and the single sample, respectively. Then we have the following covariance matrix:

$$
\Sigma(w) := \operatorname{cov}[r] = \frac{1}{n} \sum_{i=1}^{n} (r_i - \nabla f(w))(r_i - \nabla f(w))^T,
\tag{36}
$$

where $n$ indicates the total number of training samples. Likewise, a stochastic gradient $g$ computed on a randomly-drawn mini-batch is a random variable with mean $\nabla f(w)$. Assuming that it is composed of $b$ samples drawn independently with replacement, its covariance matrix is:

$$
\operatorname{cov}[g] = \frac{\Sigma(w)}{b}.
\tag{37}
$$

According to the Central Limit Theorem, g can be approximately normally distributed:

$$g \sim \mathcal{N}\left(\nabla f(w), \frac{\Sigma(w)}{b}\right).\tag{38}$$

As assumed in Appendix A.4.1 section, the variance of stochastic gradients with batch size $b_i$ meets $\mathbb{E}\left\|g^{(i)} - \nabla_i f(w)\right\|^2 \leq \sigma_i^2$ for all $w \in \mathbb{R}^d$ and $i \in [1, h]$. So when we increase the batch size from $b_i$ to $Mb_i$, we have:

$$\mathbb{E}\left\|g^{(i)} - \nabla_i f(w)\right\|^2 \leq \frac{\sigma_i^2}{M}.\tag{39}$$

By substituting $\sigma_i^2$ with $\frac{\sigma_i^2}{M}$ for all $i \in [1, h]$ in Eq.(21), we get:

$$\frac{1}{T}\sum_{t=1}^{T}\mathbb{E}\left[\|\nabla f(w_t)\|^2\right] \leq \frac{2(f(w_1) - f_{inf})}{T\eta_t} + \sum_{i=1}^{h}\eta_t L_i\left(K\frac{\sigma_1^2}{M} + (1 + \alpha_0)\frac{\sigma_i^2}{M}\right).\tag{40}$$

Let $\eta_t = \sqrt{\frac{M}{T}}$, we obtain a $O(1/\sqrt{MT})$ convergence rate.

## A.5 Parameter Settings

### A.5.1 Settings for Different Visual Predictors

In this section, we give the detailed hyper-parameter settings for the training of different visual predictors, which are shown in Table 1, Table 2 and Table 3. All predictors are evaluated on the **validation set** of COCO and ADE20K datasets. For SGD optimizer, we do not follow the linear learning rate scaling in [5] since the large learning rate on batch size 512 leads to the training failure of baseline. Instead, when the batch size is greater than 128 (256 for semantic segmentation), we use the square root of learning rate scaling to avoid divergence in the training process. With this strategy, we obtain a better baseline than [5]. Especially, the best learning rate on Faster R-CNN on batch size 512 is 0.38. For AdamW optimizer, the learning rate scaling strategy is almost the same as SGD. The only difference is that we adopt a smoother scaling scheme due to its faster convergence speed. Specifically, when the batch size is greater than 128, the learning rate is scaled up with a ratio of $\sqrt{1.5}$ if we double the batch size.

Table 1: Hyper-parameter settings for SGD optimizer on Faster R-CNN, Mask R-CNN, and Panoptic FPN with the CNN backbone. LR represents the global learning rate.

| Batch Size | Warmup Epochs | LR | LR Decay | $\tau$ | $\alpha$ | Weight Decay |
|---|---|---|---|---|---|---|
| 32 | 1 | 0.04 | MultiStep | 10 | 0.97 | 1e-4 |
| 256 | 2 | 0.226 | MultiStep | 10 | 0.97 | 1e-4 |
| 512 | 2 | 0.32 | MultiStep | 10 | 0.97 | 1e-4 |
| 1024 | 2 | 0.452 | MultiStep | 5 | 0.97 | 1e-4 |

Table 2: Hyper-parameter settings for SGD optimizer on Semantic FPN with the CNN backbone. LR represents the global learning rate. "Poly" means that the learning rate at current iteration is multiplied by $(1 - \frac{iter}{max\_iter})^{power}$ (with $power = 0.9$).

| Batch Size | Warmup Iters | LR | LR Decay | $\tau$ | $\alpha$ | Weight Decay |
|---|---|---|---|---|---|---|
| 32 | 500 | 0.01 | Poly | 5 | 0.97 | 5e-4 |
| 512 | 500 | 0.113 | Poly | 5 | 0.97 | 5e-4 |
| 1024 | 250 | 0.16 | Poly | 5 | 0.97 | 5e-4 |
| 2048 | 125 | 0.226 | Poly | 5 | 0.97 | 5e-4 |

Table 3: Hyper-parameter settings for AdamW optimizer on Faster R-CNN with the Tranformer backbone. LR represents the global learning rate.

| Batch Size | Warmup Epochs | LR | LR Decay | $\tau$ | $\alpha$ | Weight Decay | Gradient Clip |
|---|---|---|---|---|---|---|---|
| 32 | 1 | 2e-4 | MultiStep | 10 | 0.97 | 0.05 | - |
| 256 | 2 | 9.8e-4 | MultiStep | 10 | 0.97 | 0.05 | 1.0 |
| 512 | 2 | 1.2e-3 | MultiStep | 10 | 0.97 | 0.05 | 1.0 |
| 1024 | 3 | 1.5e-3 | MultiStep | 5 | 0.97 | 0.05 | 1.0 |

### A.5.2 Settings for Billion-level UniNet

Table 4: UniNet-G architecture. We adopt the Fused MBConv blocks [6] and transformer blocks to form a hybrid convolution-transformer visual network.

| Stage | Block | Network Size | | |
|---|---|---|---|---|
| | | Expansion | Channel | Layers | Stride |
| 0 | Fused MBConv | 1 | 104 | 6 | 2 |
| 1 | Fused MBConv | 4 | 216 | 9 | 4 |
| 2 | Fused MBConv | 6 | 384 | 18 | 8 |
| 3 | Fused MBConv | 3 | 576 | 18 | 16 |
| 4 | Transformer | 2 | 576 | 18 | 16 |
| 5 | Transformer | 5 | 1152 | 36 | 32 |

We scale the UniNet [7] to 1-billion parameters and evaluate it on COCO **test-dev** benchmark. The detailed architecture is presented in Table 4.

**Improved HTC detector.** To compare with the state-of-the-art, we implement some extensions to the original HTC [8] and denote it as HTC-X. This improved version is built upon the light-weight variant of HTC (HTC-Lite [9]). To reduce the computation overheads, the transformer blocks of UniNet-G backbone are evenly split into 18 subsets. There are two blocks using window attention and the last block using global attention in each subset. Furthermore, we adopt RCNet [10] and SEPC [11] as the feature pyramid with levels from $P_3$ to $P_8$, and increase the feature channel from 256 to 384. The positive IoU thresholds in the R-CNN stage are increased to 0.6, 0.7, 0.8. We use 4 decoupled transformer blocks for the classification branch and localization branch, respectively.

**ImageNet-22K pre-training.** We train the UniNet-G for 150 epochs using an AdamW optimizer and a cosine learning rate scheduler. The peak learning rate is 0.005 and the minimum learning rate is 0.0001. A batch size of 5120 and a weight decay coefficient of 0.03 are used. We adopt common augmentation techniques including Mixup, Cutmix, Random Erasing, and stochastic depth with a ratio of 0.3.

**Finetuning on COCO object detection.** We first finetune the improved HTC-X (without the mask branch) on the Objects-365 V1 dataset [12], which consists of 638k images. The model is trained with an AdamW optimizer with a learning rate of $8e-5$ and a batch size of 64 for 20 epochs. Then we further finetune it on COCO dataset for only 11 epochs. A batch size of 960 and a learning rate of $1.5e-4$ are adopted. During the finetuning phase, the shorter side of the input image is randomly selected between 400 and 1200 while the longer side is at most 1600. The window sizes of UniNet-G are set to $28 \times 28$ for Stage 4 and $14 \times 14$ for Stage 5.

### A.6 Overview of AGVM-enabled SGD and AdamW

We treat the `Backbone` ($i = 1$) as the anchor and modulate other modules making their gradient variances consistent with the `Backbone`. Specifically, we adjust the module learning rates $\hat{\eta}_t^{(i)}$ by using the ratio between $\Phi_t^{(1)}$ and $\Phi_t^{(i)}$. The update rule for each network module can be written as:

$$w_{t+1}^{(i)} = w_t^{(i)} - \hat{\eta}_t^{(i)} g_t^{(i)}, \text{ where } \hat{\eta}_t^{(i)} = \eta_t \mu_t^{(i)} \text{ and } \mu_t^{(i)} = \sqrt{\frac{\Phi_t^{(1)}}{\Phi_t^{(i)}}}, \qquad (41)$$

**Algorithm 1** AGVM+SGD

**Input:** $w_1 \in \mathbb{R}^d$, learning rate $\{\eta_t\}_{t=1}^T$, parameters $0 \le \beta_1, \alpha < 1$, interval $\tau$, weight decay coefficient $\lambda$
Set $m_0 = 0$, $u_0^{(i)} = 1$ for $i \in [1, h]$
**for** $t = 1$ **to** $T$ **do**
    Draw b samples $S_t$ from dataset $S$
    Compute $g_t = \frac{1}{b} \sum_{j \in S_t} \nabla l\left(w_t, (x_j, y_j)\right)$
    **if** $t\%\tau = 0$ **then**
        Compute $\Phi_t^{(i)}$ via gradients $g_t^{(i)}$
        Compute $\hat{\eta}_t^{(i)}$ and $\mu_t^{(i)}$
    **end if**
    $m_t = \beta_1 m_{t-1} + (1 - \beta_1)(g_t + \lambda w_t)$
    $w_{t+1}^{(i)} = w_t^{(i)} - \hat{\eta}_t^{(i)} m_t^{(i)}$
**end for**

**Algorithm 2** AGVM+AdamW

**Input:** $w_1 \in \mathbb{R}^d$, learning rate $\{\eta_t\}_{t=1}^T$, parameters $0 \le \beta_1, \beta_2, \alpha < 1$, interval $\tau$, weight decay coefficient $\lambda$
Set $m_0 = 0$, $v_0 = 0$, $u_0^{(i)} = 1$ for $i \in [1, h]$
**for** $t = 1$ **to** $T$ **do**
    Draw b samples $S_t$ from dataset $S$
    Compute $g_t = \frac{1}{b} \sum_{j \in S_t} \nabla l\left(w_t, (x_j, y_j)\right)$
    $m_t = \beta_1 m_{t-1} + (1 - \beta_1)g_t$
    $v_t = \beta_2 v_{t-1} + (1 - \beta_2)g_t^2$
    **if** $t\%\tau = 0$ **then**
        Compute $\Phi_t^{(i)}$ via modified gradients $\frac{g_t^{(i)}}{\sqrt{v_t + \epsilon}}$
        Compute $\hat{\eta}_t^{(i)}$ and $\mu_t^{(i)}$
    **end if**
    $m_t = \frac{m_t}{1 - \beta_1^t}$, $v_t = \frac{v_t}{1 - \beta_2^t}$, $r_t = \frac{m_t}{\sqrt{v_t + \epsilon}}$
    $w_{t+1}^{(i)} = w_t^{(i)} - \hat{\eta}_t^{(i)}(r_t^{(i)} + \lambda w_t^{(i)})$
**end for**

where $\eta_t$ is the global learning rate. However, simply adjusting the learning rates on-the-fly would easily yield training failure due to the transitory large variance ratio that impedes the optimization. We propose a momentum update to address this problem. Let $\alpha \in [0, 1)$ be a momentum coefficient, we have:

$$\mu_t^{(i)} \leftarrow \alpha \mu_{t-1}^{(i)} + (1 - \alpha)\mu_t^{(i)}, \tag{42}$$

which can reduce the influence of unstable variance. Note that we update $\mu_t^{(i)}$ each $\tau$ iterations. Based on this, we present AGVM-enabled SGD and AdamW optimizers in Alg. 1, and Alg. 2. In the practical implementation in extremely-large batch regime (*e.g.,* 10k), we add a small epsilon value $\mu_t^{(i)} = \sqrt{\frac{\Phi_t^{(1)} + \epsilon}{\Phi_t^{(i)} + \epsilon}}$ in Eq.(41) to ensure stability and also clip the $\mu_t^{(i)}$ to [0.1, 10].

## A.7 Related Work on Dense Visual Predictions

We can divide current deep learning based object detection into two-stage and single-stage detectors. A network that has a separate module to generate region proposals is termed as a two-stage detector. These methods try to find an arbitrary number of proposals in an image during the first stage and then classify and localize them in the second stage, including Faster R-CNN [3], Mask R-CNN [13], and R-FCN [14]. Single-stage detectors, such as SSD [15] and RetinaNet [16], classify and localize semantic objects in a single shot using dense sampling. They use predefined boxes/keypoints of various scales and aspect ratios to localize objects. Some single-stage detectors, like FOCS [17] can also achieve competitive results with two-stage detectors. In recent years, deep learning models have yielded a new generation of image segmentation [18, 19] tasks with significant performance improvements. Different from detection tasks, we can group deep learning segmentation based on the segmentation goal into semantic segmentation, instance segmentation, and panoptic segmentation. Semantic segmentation [20, 21] can be seen as an extension of image classification from image level to pixel level, while instance segmentation [13, 22] can be defined as the task of finding simultaneous solution to semantic segmentation and object detection. Finally, panoptic segmentation [23, 24, 25] focus on identifying things and stuff separately, also separating (using different colors) the things of the same class.