# OpenReview forum: "Large-batch Optimization for Dense Visual Predictions: Training Faster R-CNN in 4.2 Minutes"
_NeurIPS.cc/2022/Conference — NeurIPS 2022 Accept_

### Official Review · Reviewer_q5yJ · 2022-07-07

**Rating:** 5
**Confidence:** 3
**Soundness:** 3 good
**Presentation:** 3 good
**Contribution:** 2 fair

**Summary:**

The paper presents a new approach called Adaptive Gradient Variance Modulator （AGVM）for large batch-size training. The approach is well motivated with simple implementation. Experimental results on different kinds of dense prediction tasks like object detection, instance segmentation, panoptic segmentation validate the effectiveness of the approach. More specifically, the paper claims to be able to train with a batchsize of 10K.

**Questions:**

1. According to Table 2, the results with batch-size 1024 (and 512) are usually lower than the setting with batchsize of 32 (or 256), expecially for the case of object detection 35.4 vs 36.8. The performance loss is not negligible with large-batchsize setting.

2. In the abstract, the paper claims that the batchsize can be scalable to 10K. But the experiments in Table 5 cannot fully support the claim. First, the training setting is compromised with small backbone like Resnet18. How about the performance of the Res50 backbone, which is widely used in other experimental setting? Also, how about other detection methods rather than RetinaNet. Moreover, the reported results of 10K batchsize is much lower than batch-size of 32.

3. For the results discussed in Table 4, only training time is reported without the converence results. Also, when more GPUs have been adopted, the accelerator is a little bit lower. For example, the training time is 4.2 mins with 1536 GPUs vs 148 mins with 32 GPUs.

**Limitations:**

The authors addresed the limitations when the batchsize cannot be well estimated effectively like heatmap based pose estimation. It also addressed the negative social impact when used for deepfake training.

One limitation which has not reported is that the performance drop when large batch-size has been utilized.

**Strengths And Weaknesses:**

strengths:
1. The proposed approach for large batchsize training is simple and easy to implement.
2. The proposed AGVM can be generally applied to different visual prediction tasks like instance segmentation, panoptic segmentation.


weakness:
1. Training with large batchsize with the proposed approach leads to performance drop. The performance gap is obvious if larger batch-size is adopted.
2. The speed-up ratio is not constant with more GPUs are utilized.

---

> ### Author Response · Authors · 2022-08-02
> **Response to Reviewer q5yJ**
>
> Dear Reviewer q5yJ
>
> Thank you for the detailed review. We will address your concerns below.
>
> **Q1:The performance gap is obvious if a larger batch-size is adopted, especially for object detection task in Table 2**
>
> Thanks for this question. As large-scale dataset and big models become democratized, shortening the training time is very important for exploring this. Large-batch training is a promising way to achieve this and it's becoming a very challenging research field.
>
> Recently, the whole research community is working on eliminating the performance drop in large-batch optimization [1][2][3][4][5][6][7]. They concentrate on the basic image classification task and a specific task in dense visual predictions, i.e., object detection. Even though great effort has been made by the researchers, the performance loss still exists in large-batch setting. Theoretically, under fixed computation complexity (i.e., same training epochs),  this is **inevitable** for performance drop when batch size is very large [8].
>
> We argue that we contribute two main advances to the community. One is that we are the first to extend the large-batch optimization problem to dense visual predictions rather than being limited to a specific object detection task. We further reveal that the essential reason for this optimization problem is the significant effective batch size misalignment between different modules. The other is we propose a generalized AGVM which shows overwhelming superiority over all of the previous methods. **It establishes many new stage-of-the-art performances for dense visual prediction tasks (more than twenty) and improves the maximum batch size that the algorithm works without performance loss**.
>
> **Q2: Why ResNet18?  How about other pipelines other than RetinaNet?**
>
> Thanks for your constructive comment.
> We further conduct experiments with ResNet50 backbone based on RetinaNet and Faster R-CNN.
>
> **RetinaNet+ResNet50 (2x)**
>
> |Batch Size| 32 |256 | 1k | 2k | 4k |10k |
> |----------|----|----|----|----|----|----|
> | PMD-LAMB | 36.5   | 36.6   | 34.8   | 31.5   |27.1|NaN|
> | AGVM     | 37.1   | 37.1   |36.7   | 34.1   | 33.0   | 30.8   |
>
> **Faster R-CNN+ResNet50 (1x)**
>
> |Batch Size| 32 | 256 | 1024 | 1536 |  2k  |  4k |
> |----------|----|------|------|------|------|-----|
> | PMD-LAMB |36.6| 36.7 | 35.3 | 33.5 | 28.7 | NaN |
> | AGVM     |37.1| 37.2 | 37.0 | 36.6 | 33.8 | 25.2|
>
> **Q3: The reported results of 10K batchsize is much lower than batch-size of 32**
>
> Sorry for this confusing description that the batchsize can be scalable to 10K (this doesn't refer to the batch size without performance drop), we will clarify this below. The evaluation on large-batch training in the research community can be divided into two aspects, **the maximum batch size that the algorithm works without performance loss** and **the extreme batch size that the algorithm converges without NaN**.
>
> **RetinaNet+ResNet50**
>
> For the maximum batch size that the algorithm works without performance loss, we list the detailed comparison where it requires that the algorithm can achieve 36.6 mAP@0.5:0.95 within 24 epochs.
>
> | Method    | MegDet | LAMB | PMD-LAMB | LARS | AGVM |
> |-----------|--------|------|----------|------|------|
> | Batch size | 256    | 128  | 256      | 128  | 1k   |
>
> For the extreme batch size that the algorithm converges without NaN, we conduct multiple experiments to evaluate this.
>
> | Method                        | MegDet | LAMB | PMD-LAMB | LARS | AGVM |
> |-------------------------------|--------|------|----------|------|------|
> | Batch size | 2k (NaN)    | 6k (NaN)  | 6k (NaN)      | 2k (NaN)  | 10k (30.8) |
>
> Note that even with 10k batch size, AGVM still performs well convergence and due to the limitation of GPU resources, we could not explore a larger batch size.
>
> **Faster R-CNN+ResNet50**
>
> For the maximum batch size that the algorithm works without performance loss, we list the detailed comparison where it requires that the algorithm can achieve 36.6 mAP@0.5:0.95 within 12 epochs (defined in [7]).
>
> | Method    | MegDet | LAMB | PMD-LAMB | LARS | AGVM |
> |-----------|--------|------|----------|------|------|
> | Batch size | 128    | 128  | 320      | 128  | 1.5k   |
>
> For the extreme batch size that the algorithm converges without NaN, we conduct multiple experiments to evaluate this.
>
> | Method                        | MegDet | LAMB | PMD-LAMB | LARS | AGVM |
> |-------------------------------|--------|------|----------|------|------|
> |Batch size | 2k (NaN)     | 4k(NaN)   | 4k (NaN)       | 2k (NaN)   | 4k (25.2)   |
>
> AGVM demonstrates consistent superiority over prior arts in these two aspects.

---

> > ### Author Response · Authors · 2022-08-02
> > **Response to Reviewer q5yJ**
> >
> > **Q4: When more GPUs have been adopted, the accelerator is a little bit lower**
> >
> > Ideally, increasing the batch size will linearly decrease the training iterations and so we can shorten the training time linearly. However, in practice, it is difficult for us to achieve this ideal state. More specifically, it requires more GPUs to achieve a larger batch size which increases the communication burden for many necessary synchronization operations such as gradient synchronization between nodes after back-propagation. This overhead is influenced by the GPU numbers and the cluster topology, which directly leads to that we can't achieve 48 times speed-up when increasing 48 times batch size and adopting 48 times computational resources.
> >
> > **In detail, for Faster R-CNN, a gradient synchronization operation takes less than 5ms with 16 GPUs, but takes 20-40ms with 768 GPUs at each iteration because of the barrier synchronization.** Therefore, the gradient synchronization operation is the bottleneck for imperfect speedup. And distributed deep learning framework also will influence the system throughput when increasing GPUs. It's also impractical to achieve linear speedup in Google TPU clusters (see Table 1 in [2])
> >
> > To sum up, the reasons for the imperfect speedup are mainly related to cluster performance, not AGVM. AGVM only introduces a negligible extra overhead compared to the regular set-up (see the response to Reviewer 8EAD). You can view the detailed reasons for imperfect speedup for UniNet-G in the response to Q2 from Reviewer 8KtA.
> >
> > **Q5: Convergence results in Table 4**
> >
> > Thanks for this comment. In a nutshell, all of the convergence results in Table 4 are 36.6 mAP@0.5:0.95 for Faster R-CNN+ResNet50.
> > In section 4.1, we have demonstrated that we explore how fast AGVM can reach the **36.6 mAP@0.5:0.95** reported in [7] to make a fair comparison with [7] (PMD-LAMB, which needs 12 minutes to train). We follow the same settings with [7] and report the training time in Table 4. We reduce the original small-batch training time from 2.5 hours to only 4.2 minutes, which is the fastest record to our knowledge.
> >
> > [1] You Y, Zhang Z, Hsieh C J, et al. Imagenet training in minutes[C]//Proceedings of the 47th International Conference on Parallel Processing. 2018: 1-10.
> >
> > [2] You Y, Li J, Reddi S, et al. Large batch optimization for deep learning: Training bert in 76 minutes[J]. arXiv preprint arXiv:1904.00962, 2019.
> >
> > [3] Goyal P, Dollár P, Girshick R, et al. Accurate, large minibatch sgd: Training imagenet in 1 hour. arXiv 2017[J]. arXiv preprint arXiv:1706.02677, 2017.
> >
> > [4] Liu Y, Mai S, Chen X, et al. Towards efficient and scalable sharpness-aware minimization[C]//Proceedings of the IEEE/CVF Conference on Computer Vision and Pattern Recognition. 2022: 12360-12370.
> >
> > [5] Liu Y, Chen X, Cheng M, et al. Concurrent adversarial learning for large-batch training[J]. arXiv preprint arXiv:2106.00221, 2021.
> >
> > [6] You, Yang, et al. "The limit of the batch size." arXiv preprint arXiv:2006.08517 (2020).
> >
> > [7] Wang T, Zhu Y, Zhao C, et al. Large Batch Optimization for Object Detection: Training COCO in 12 minutes[C]//European Conference on Computer Vision. Springer, Cham, 2020: 481-496.
> >
> > [8] Zhao S Y, Xie Y P, Li W J. Stochastic Normalized Gradient Descent with Momentum for Large Batch Training[J]. arXiv preprint arXiv:2007.13985, 2020.

---

> ### Author Response · Authors · 2022-08-08
> **Looking forward to your post-rebuttal feedback**
>
> Dear Reviewer q5yJ:
>
> We thank you for the precious review time and valuable comments. We have provided corresponding responses and results, which we believe have covered your concerns. We hope to further discuss with you whether or not your concerns have been addressed. Please let us know if you still have any unclear parts of our work.  If your concerns have been well addressed, please consider raising your rating, thanks.
>
> Best,
>
> Authors

---

### Official Review · Reviewer_8KtA · 2022-07-10

**Rating:** 4
**Confidence:** 3
**Soundness:** 3 good
**Presentation:** 3 good
**Contribution:** 3 good

**Summary:**

This paper proposes an Adaptive Gradient Variance Modulator (AGVM) to achieve large-batch optimization for dense visual tasks, i.e., object detection and segmentation. The paper clearly claims the motivation, which is the high gradient variance in the different modules in large-batch dense visual tasks. To overcome the issue in large-batch dense visual tasks, the authors propose the AGVM to make the gradient variance of different modules consistent and moving average to have stable optimization. The AGVM has a theoretical guarantee, and sufficient experiments have also proved its effectiveness.

**Questions:**

See the weakness. I will consider raising the score if the authors address the above issues.

**Limitations:**

Yes.

**Strengths And Weaknesses:**

Strength
1. The paper is clear and well written. The paper clearly studies the difference between dense visual tasks and normal classification tasks.
2. The method AGVM proposed has a theoretical guarantee.
3. The method is evaluated in different tasks and different optimizers. The experiments show comparable or better performance compared to the previous state-of-the-art optimization.


Weakness
1. The Equ(4) shows the AGVM uses Backbone as the anchor, so the ratio is \sqrt{\frac{\Phi^1}{\Phi^i}}. Why choose backbone? Will using other modules lead to training failure?
2. Table 8 shows the results in different batch sizes. What are the experimental results of other batch sizes, e.g., 128, 512, and the batch size increased 30 times from 32 to 960, but the training time was shortened 20 times, why does AGVM not achieve linear speed? If the bottleneck is data loader IO, please report its time.
3. Table 3 shows the performance in different batch sizes with different optimizers. We wonder about the results of AdamW since the AdamW works well in small batch sizes, i.e., 16, 32.
4. The paper claims that the high gradient variance is the reason for the failure of large-batch dense visual training. It’s recommended to report the same variance-iteration figure as the Figure1 with AGVM to complete the paper.
5. Recently, the work [r1] also study the large-scale DNN training from the perspective of gradient variance misalignment. The authors should give a fair comparison and discussion on this related work.
6. About theoretical analysis.  Can the authors provide a rigorous guarantee for Eq (7) in the appendix, which is merely numerical observation rather than a theoretical analysis?
7. The theoretical contribution is mild. We recommend the authors provide the linear speedup property of the proposed AGVM with respect to the number of works or mini-batch size.

[r1] Ko, Yunyong, Dongwon Lee, and Sang-Wook Kim. "Not All Layers Are Equal: A Layer-Wise Adaptive Approach Toward Large-Scale DNN Training." Proceedings of the ACM Web Conference 2022. 2022.

---

> ### Author Response · Authors · 2022-08-02
> **Response to Reviewer 8KtA**
>
> We thank the reviewer for taking the time to review our paper and give the point-to-point response below. We really hope our response addresses your concerns. If so, please consider raising your rating, thanks.
>
> **Q1: Why choose backbone as the anchor module?**
>
> The significant **effective batch size misalignment** between different sub-modules as we investigated in section 4.2 is the essential reason for unstable large-batch training. This eventually leads to the imbalanced gradient variance in different modules. Intuitively, the effective batch size of the backbone is equal to the actual input batch size. In contrast, the effective batch size of the other modules is difficult to quantify and from our observation, as shown in Figure 1, they are also more volatile under different batch sizes.
>
> We further conduct experiments to evaluate this. As shown in Table 7 in our paper (Mask R-CNN with batch size 512), we choose different modules as the anchor and we can see that adopting the backbone as the anchor leads to the best performance.
>
> **Q2: UniNet-G performances and why not a linear speed-up ratio. If the bottleneck is data loader IO, please report its time.**
>
> Thanks for your suggestion. We first give the performance of AGVM with batch sizes 128 and 512:
>
> | batch size | method | Box mAP | Seg mAP | Iterations | Training time |
> | ---------- | ------ | ------- | ------- | ---------- | ------------- |
> | 128        | AGVM   | 62.6    | 53.8    | 11004      | 21.3 hours    |
> | 128        | AdamW  | 62.5    | 53.7    | 11004      | 21.1 hours    |
> | 512        | AGVM   | 62.5    | 53.7    | 2760       | 5.9 hours     |
> | 512        | AdamW  | 61.8    | 53.0    | 2760       | 5.8 hours     |
>
> **Why not linear speed-up**:
>
> Theoretically, increasing the batch size will linearly decrease the training iterations and so we can shorten the training time linearly.
> However, in practice, it is difficult for us to achieve this ideal state. More specifically, it requires more GPUs to achieve a larger batch size, which increases the communication burden for many necessary synchronization operations such as gradient synchronization between nodes after back-propagation. This overhead is influenced by the GPU numbers and the cluster topology, which directly leads to that we can't achieve 30 times speed-up when increasing 30 times batch size and adopting 30 times computational resources.
>
> **In detail, for Uninet-G, a gradient synchronization operation takes less than 0.3s with 16 GPUs, but takes 2-2.5s with 480 GPUs at each iteration because of the barrier.** Therefore, the gradient synchronization operation is the bottleneck for imperfect speedup rather than Dataloader IO (cost less than 50ms). Furthermore, the overhead of an all-reduce call will increase with the number of GPUs for UniNet-G. An all-reduce call takes 68.9ms with 16 GPUs, but takes 148.1ms with 480 GPUs. And distributed deep learning framework will also influence the system throughput when increasing GPUs. It's also impractical to achieve linear speedup in Google TPU clusters (see Table 1 in [1]).
>
> To sum up, the reasons for the imperfect speedup are mainly related to cluster performance, not AGVM. AGVM only introduces a negligible extra overhead compared to the regular set-up. (see the response to Reviewer 8EAD Q2)
>
> **Q3:  Results with AdamW**
>
> Thanks for your comment. We list the results in the following:
> Faster R-CNN + ResNet50
>
> | Batch size | AdamW | AGVM+AdamW | Iterations |
> | ---------- | ----- | ---------- | ---------- |
> | 16         | 37.1  | 37.2       | 87960      |
> | 32         | 37.1  | 37.1       | 43980      |
> | 256        | 36.9  | 37.2       | 5508       |
> | 512        | 36.2  | 36.8       | 2760       |
> | 1024       | 36.2  | 37.0       | 1380       |
> | 1536       | 35.9  | 36.6       | 924        |
>
> Faster R-CNN + Swin-Tiny
>
> | Batch size | AdamW | AGVM+AdamW | Iterations |
> | ---------- | ----- | ---------- | ---------- |
> | 16         | 43.7  | 43.7       | 95350      |
> | 32         | 43.6  | 43.7       | 47675      |
> | 256        | 43.4  | 43.5       | 5967       |
> | 512        | 42.7  | 43.2       | 2990       |
> | 1024       | 42.4  | 42.8       | 1495       |
>
> AGVM demonstrates consistent superiority over basic AdamW with different batch sizes.
>
> **Q4: The Same Variance-iteration Figure as The Figure 1 with AGVM**
>
> Thanks for your constructive comment. We attach the results in Appendix Figure 3 in the rebuttal version.

---

> > ### Author Response · Authors · 2022-08-02
> > **Response to Reviewer 8KtA**
> >
> > **Q5: Comparisons with LENA [2]**
> >
> >  Lena adopts the "gradient variance" from the "gain ratio" in AdaScale [3] to modify layer-wise learning adaptively. The differences between LENA and AGVM are given in the following:
> >
> > 1. LENA keeps increasing the learning rate in the whole training process, resulting in a very unstable training for object detection and segmentation.
> > 2. AGVM is motivated by effective batch size misalignment in dense visual predictions to modify module-wise learning rate. But LENA focuses on image classification problems and modifies layer-wise learning rate.
> > 3. LENA cannot generalize well to dense visual prediction tasks such as object detection. We give the experiment results with Faster R-CNN+ResNet50 detector on COCO. We implement LENA by borrowing its official implementation (https://github.com/yy-ko/lena-www22) and adopt the recommended $\theta$ and $\alpha$, but we cannot obtain any reasonable performance：
> >
> > | Batch size | LENA | AGVM |
> > |------------|------|------|
> > | 256        | 32.5 | 36.7 |
> > | 512        | 25.9 | 36.7 |
> > | 1024       | 18.6 | 35.4 |
> >
> > 4. LENA needs an experienced engineer to tune the hyper-parameters, while AGVM does not.
> >
> > **Q6: A rigorous guarantee for Eq (7) (Eq.(12) in the rebuttal version) in the appendix**
> >
> > Thanks for your constructive comment. Because the samples are randomly divided into two groups, according to the law of large numbers, when batch size $b$ goes to infinity, we have:
> >
> > $$
> > \mathbb{E}\left[cos(G_{t,1}^{(j)},G_{t,2}^{(j)})\right] \to 1, \forall j\geq 1.
> > $$
> >
> > For $b=2$, each group only has one sample that comes from the same training distribution, we have:
> >
> > $$
> > \mathbb{E}\left[cos(G_{t,1}^{(j)},G_{t,2}^{(j)})\right] \to 0, \forall j\geq 1.
> > $$
> >
> > Therefore, there exists a $\hat{b}$ that makes the following equation hold,
> >
> > $$
> > \mathbb{E}\left[cos(G_{t,1}^{(j)},G_{t,2}^{(j)})\right] \leq \frac{1}{2}, {\rm if}\, b\leq \hat{b}, \forall j\geq 1.
> > $$
> >
> > Since the effective batch size of backbone is smaller than that of other modules, the gradient variance of backbone is larger than that of other modules, which means:
> >
> > $$
> > \mathbb{E}\left[cos(G_{t,1}^{(1)},G_{t,2}^{(1)})\right] \leq \mathbb{E}\left[cos(G_{t,1}^{(i)},G_{t,2}^{(i)})\right], \forall i > 1.
> > $$
> >
> > When $b<\hat{b}$, we further have:
> >
> > $$
> > \mathbb{E}\left[cos(G_{t,1}^{(1)},G_{t,2}^{(1)})\right](1 - \mathbb{E}\left[cos(G_{t,1}^{(1)},G_{t,2}^{(1)})\right]) \leq \mathbb{E}\left[cos(G_{t,1}^{(i)},G_{t,2}^{(i)})\right](1 - \mathbb{E}\left[cos(G_{t,1}^{(i)},G_{t,2}^{(i)})\right]), \forall i > 1.
> > $$
> >
> > Then we get the Eq.(7).
> >
> > **Q7: Linear speedup property of the proposed AGVM**
> >
> > Thanks for your suggestion. We give the linear speedup property for AGVM+synchronous SGD w.r.t. batch size as a corollary.  First, we will prove **gradient variance decreases linearly with batch size b**. For ease of understanding, we assume that $\nabla f(w)$, $g$, $r$ represent the gradient of the full dataset, the mini-batch with size $b$ and the single sample, respectively. Then we have the following covariance matrix:
> >
> > $$
> > \Sigma(w):=\operatorname{cov}\left[r\right]=
> > \frac{1}{n} \sum_{i=1}^{n}\left(r_i-\nabla f(w)\right)\left(r_i-\nabla f(w)\right)^{T},
> > $$
> >
> > where n indicates the total number of training samples. Likewise, a stochastic gradient $g$ computed on a randomly-drawn mini-batch is a random variable with mean $\nabla f(w)$. Assuming that it is composed of $b$ samples drawn independently with replacement, its covariance matrix is:
> >
> > $$
> > \operatorname{cov}[g]=\frac{\Sigma(w)}{b}.
> > $$
> >
> > According to the Central Limit Theorem, g can be approximately normally distributed:
> >
> > $$
> > g \sim \mathcal{N}\left(\nabla f(w), \frac{\Sigma(w)}{b}\right).
> > $$
> >
> > As assumed in Appendix A.4.1 section, the variance of stochastic gradients with batch size $b_i$ meets $\mathbb{E}\left\|g^{(i)}-\nabla_{i} f(w)\right\|^{2} \leq \sigma_{i}^{2}$ for all $w \in \mathbb{R}^{d}$ and $i \in[1,h]$.
> > So when we increase the batch size from $b_i$ to $Mb_i$, we have:
> >
> > $$
> > \mathbb{E}\left\|g^{(i)}-\nabla_{i} f(w)\right\|^{2} \leq \frac{\sigma_{i}^{2}}{M}.
> > $$
> >
> > By substituting $\sigma_{i}^{2}$ with $ \frac{\sigma_{i}^{2}}{M}$ for all $i \in [1,h]$, we get:
> >
> > $$
> > \frac{1}{T}\sum_{t=1}^{T}\mathbb{E}\left[ \|\nabla f\left(w_{t}\right)\|^{2}\right]\leq \frac{2\left(f\left(w_{1}\right)-f_{inf}\right)}{T\eta_{t}}+\sum_{i=1}^{h}\eta_{t} L_{i}\left(K\frac{\sigma_{1}^{2}}{M}+(1+\alpha_{0})\frac{\sigma_{i}^{2}}{M}\right).
> > $$
> >
> > Let $\eta_{t}=\sqrt{\frac{M}{T}}$, we obtain a $O(1/\sqrt{MT})$ convergence rate.
> >
> > [1] You Y, Li J, Reddi S, et al. Large batch optimization for deep learning: Training bert in 76 minutes[J]. ICLR, 2020.
> >
> > [2] Ko Y, Lee D, Kim S W. Not All Layers Are Equal: A Layer-Wise Adaptive Approach Toward Large-Scale DNN Training[C]//Proceedings of the ACM Web Conference 2022. 2022: 1851-1859.
> >
> > [3] Johnson T, Agrawal P, Gu H, et al. AdaScale SGD: A user-friendly algorithm for distributed training[C]//International Conference on Machine Learning. PMLR, 2020: 4911-4920.

---

> ### Author Response · Authors · 2022-08-08
> **Looking forward to your post-rebuttal feedback**
>
> Dear Reviewer 8KtA:
>
> We thank you for the precious review time and valuable comments. We have provided corresponding responses and results, which we believe have covered your concerns. We hope to further discuss with you whether or not your concerns have been addressed. Please let us know if you still have any unclear parts of our work. If your concerns have been well addressed, please consider raising your rating, thanks.
>
> Best,
>
> Authors

---

> > ### Comment · Reviewer_8KtA · 2022-08-09
> > **Post-rebuttal discussion**
> >
> > Thanks for the authors' detailed response.  Most of my concern has been solved. However, the reviewer still has the following concerns:
> >
> > (1) The rigorous guarantee for Eq (7) is still not convincing.  I recommend the authors estimate the variance term with respect to the minibatch size $b$ and the iteration  $T$.  Based on the current analysis in the response, the variance term is related to an unknown mini-batch size $b$,    which may appear in the upper bound of the Theorem. Thus, the linear speedup property may be broken.  In addition, a similar issue arose in the analysis of AVGM+Adam, $1-\beta_2$ is a constant rather than approaching to 0, which is the key difficulty in estimating the convergence of Adam-type methods.
> >
> > Thus, I highly recommend the authors provide a rigorous estimation for the variance term, which will make the theoretical contribution more clear and solid.
> >
> > (2) The variance visualization in Fig 3( in appendix) does not convince me. Compared with Fig 1 (main file) and Fig 3 (appendix), the variance gap still exists between each module, which is almost the same.  Hence, the motivation "the gradient variance misalignment" in this work is questionable.
> >
> > Based on the current response, I will keep my initial score. Hope the authors can provide a more rigorous proof and make the motivation clearer.

---

> > > ### Author Response · Authors · 2022-08-09
> > > **Response to Reviewer 8KtA**
> > >
> > > **Q1: Estimate the variance term with respect to the mini-batch size b and iteration T to make the theoretical contribution more clear and solid**
> > >
> > > Thanks for your sincere comments on the theoretical proof. We would like to clarify the contributions of our work to hope that reviewer can evaluate our paper from the **core contribution** of our paper.
> > >
> > > We focus on solving a challenging problem in large-scale vision system. Our purpose is to contribute a new large-batch training algorithm for computer vision practical applications, not a theoretical algorithm to explore how to prove its' convergence very rigorously. Furthermore, our extensive empirical experiments have demonstrated the convergence and effectiveness of AGVM. Similar to previous works published on top machine learning conference [1,2,3,4], the theoretical analysis is only a **minor** insight and help further understand the properties of the method.
> > >
> > > The dense visual prediction tasks such as object detection, instance segmentation and semantic segmentation, are significantly challenging in practical applications due to the large-scale datasets and time-consuming training. Increasing the batch size by adding the GPU resources is an efficient manner to reduce the training time. However, this suffers from poor generalization issues under the large-batch training scenario. Improving the stability and scalability of large-batch optimization is an essential and significant topic for dense visual prediction tasks in many practical applications, e.g., computer vision in smart city and visual robots. Thus we propose a novel and effective large-batch optimization method AGVM for various dense prediction tasks in this paper, which shows overwhelming superiority over all of the previous state-of-the-art methods.
> > >
> > > As shown by Reviewer 8EAD's opinions, he is strongly in favor of paper acceptance and thinks that we propose a practically useful solution for large-batch training, which could help "democratize" training in several computer vision problems. Therefore, we hope that you can compare our works with the previous methods in this research field and re-examine our work in terms of its practical value and its contribution to the research community.
> > >
> > > **Last, NeurIPS conference has the following acceptance standard on machine vision: Novelty of algorithm/application, Difficulty of application, Quality of results, Insight conveyed, and Rigorous empirical evaluation (https://nips.cc/Conferences/2016/PaperInformation/EvaluationCriteria). Specifically, a NeurIPS paper on machine vision should propose a machine learning algorithm or system that can be used by a computer vision researcher to help solve a difficult computer vision problem. We firmly believe our paper and contribution deserve a positive score at least according to the NeurIPS standard. If our paper is accepted, we will definitely open source our codes.**
> > >
> > > [1] You Y, Li J, Reddi S, et al. Large batch optimization for deep learning: Training bert in 76 minutes. ICLR 2020
> > >
> > > [2] Liu Y, Chen X, Cheng M, et al. Concurrent adversarial learning for large-batch training. ICLR 2022
> > >
> > > [3] Qin H, et al. SimiGrad: Fine-Grained Adaptive Batching for Large Scale Training using Gradient Similarity Measurement. NeurIPS 2021
> > >
> > > [4] Keskar N, et al. On large-batch training for deep learning: Generalization gap and sharp minima. ICLR 2017
> > >
> > >
> > > **Q2:The variance visualization in Fig 3 (in appendix) does not convince me.**
> > >
> > > Sorry for the confusing figure. We have uploaded another version. In the previous appendix figure 3, we give the variances $\mathrm{Var} (g_{t}^{(i)})$ of the gradient $g_{t}^{(i)}$ to show AGVM could avoid failure training. But we don't add the AGVM coefficient in the previous figure 3.
> > >
> > >
> > > With AGVM, we use the modified gradient $\mu_{t}^{(i)} g_{t}^{(i)}$ to update the parameters and balance gradient variances. In current appendix figure 3, we plot the $\mathrm{Var} (\mu_{t}^{(i)} g_{t}^{(i)})$, where $\mu_{t}^{(i)}$ is the coefficient of AGVM. Then the variances have been balanced.

---

### Official Review · Reviewer_KpQK · 2022-07-11

**Rating:** 6
**Confidence:** 3
**Soundness:** 3 good
**Presentation:** 3 good
**Contribution:** 3 good

**Summary:**

This work aims to the heavy performance drop that takes place in the large-batch training regime for object detection and segmentation. Specifically, the authors propose an algorithm named Adaptive Gradient Variance Modulator (AGVM) that is able to work with a very large batch size. This work provides extensive experiments on experiments on MS COCO and ADE20K, which verify the superiority of the proposed method.

**Questions:**

Please refer to the weaknesses. I will check other reviewers' comments and the authors' responses. I will adjust my rating accordingly.

**Limitations:**

It seems that the authors addressed the limitations and potential negative societal impact.

**Strengths And Weaknesses:**

Strengths:

+ The paper is easy to follow and well-structured.

+ The experiments are well executed. They show that the proposed method can work with different deep learning backbones (e.g., CNNs and Transformers) and different optimization methods on MS COCO and ADE20K.

Weaknesses:

- It seems that $\mu_{t}^{(i)}$ is sensitive to $\Phi_{t}^{(i)}$ in Equation (4). When $G_{t,1}^{i}$ is similar to $G_{t,2}^{i}$, the expectation of the cosine of the two groups of the gradient estimation is close 0. Then $\Phi$ will be a large value.

- According to Section 3, it seems that the proposed method is generic to most computer vision tasks, like image classification. Why does the proposed method target the object detection and segmentation tasks? Can the proposed method generalize to the image classification task?

- The paper may lack a discussion about if the proposed method is able to generalize to the imbalance problems in object detection [r1].

References:
[r1] Oksuz, Kemal, et al. "Imbalance problems in object detection: A review." IEEE transactions on pattern analysis and machine intelligence 43.10 (2020): 3388-3415.

---

> ### Author Response · Authors · 2022-08-02
> **Response to Reviewer KpQK**
>
> Dear Reviewer KpQK,
>
> Thanks for your advice. We will address your concerns below.
>
> **Q1: $\mu_{t}^{(i)}$ is sensitive to $\Phi_{t}^{(i)}$. $\Phi$ will be a large value when $G_{t,1}^{(i)}$ and $G_{t,2}^{(i)}$ are similar**
>
> Thanks for this valuable question. You may question $\mu_{t}^{(i)}$ will be a large value rather than $\Phi$, since $\Phi$ is bounded between 0 and 2 in practice.
>
> Theoretically, when the batch size is extremely large (e.g., full batch), $G_{t,1}^{(i)}$ and $G_{t,2}^{(i)}$ will be similar and the gradient variance $\Phi_{t}^{(i)}$ and $\Phi_{t}^{(1)}$ tend to be zero ($\Phi_{t}^{(1)}$ will be  slightly larger than $\Phi_{t}^{(i)}$). This will lead to the unstable $\mu_{t}^{(i)}$.
>
> However, in practice, we find it's hard to achieve this. As demonstrated in **Appendix Figure 2**, even with a very large batch size 10k (requires 1280 GPUs with batch size 8 for a single GPU), the $\mu_{t}^{(i)}$ is still controllable and AGVM also has an appealing convergence property.
>
> In the practical implementation, we add a small epsilon value $\mu_{t}^{(i)}=\sqrt{\frac{\Phi_{t}^{(1)}+\epsilon}{\Phi_{t}^{(i)}+\epsilon}}$ in Eq.(4) to avoid the large value and also clip the $\mu_{t}^{(i)}$ to [0.1, 10]. We will emphasize these strategies in the revision.
>
> In addition, at the training stage, there will be some unpredictable instantaneous large $\mu_{t}^{(i)}$. To alleviate this, we further introduce a momentum update in Eq. (5) to reduce the influence of unstable $\mu_{t}^{(i)}$ in some accidental iteration.
>
> **Q2: Why does the proposed method target the object detection and segmentation tasks? Can the proposed method generalize to the image classification task?**
>
> Specifically, AGVM is proposed to alleviate the large-batch training problem. Whether it works for a given task depends on two facts, one is that this given task contains multiple sub-modules, and the other is that there is significant **effective batch size misalignment** between different sub-modules as we investigated in section 4.2, which is the essential reason for the inconsistent gradient variance.
>
> The feature pyramid network (FPN), region proposals, and shared head between different levels are the core design ideas in dense visual prediction tasks. We find this inevitable introduces the **effective batch size misalignment**, leading to the inconsistent gradient variances of different modules. Based on this, AGVM is proposed to alleviate this inconsistency.
>
> The pipeline used in the image classification task is often a single module (only backbone to directly predict the class-aware probability) which is not satisfied with these two facts above. Therefore, the application scenario of AGVM method is dense visual prediction tasks.

---

> > ### Author Response · Authors · 2022-08-02
> > **Response to Reviewer KpQK**
> >
> > **Q3: Lack of discussions on imbalanced object detection tasks.**
> >
> > Thanks for your constructive comment.
> >
> > The purpose of AGVM is to solve the large-batch training problem. The core problem it solved is that the variances of GD and SGD are not balanced between different modules due to the significant **effective batch size misalignment** as demonstrated in section 4.2. This is also a form of imbalance problem under large-batch training in dense visual prediction tasks.
> > As concluded in [1], there are other various imbalance problems in object detection such as Fg-Bg class imbalance, objective imbalance, scale imbalance, spatial imbalance, and Fg-Fg class imbalance. The essence of these imbalances is quite different from that resolved by AGVM. Thus, AGVM is specifically for large-batch optimization and can't directly solve these inherent imbalance problems in object detection.
> > We compare the essential reasons for these imbalance problems and the imbalanced gradient variance in large-batch dense visual predictions.
> >
> > | Imbalance problem     | Key reason                                   | Core solution                                 | Method                             |
> > |-----------------------|----------------------------------------------|-----------------------------------------------|------------------------------------|
> > | Fg-Bg class imbalance | Multiple negative classes (Background)       | Soft/hard sampling, Generative methods        | Focal loss, OHEM, GHM              |
> > | Objective imbalance   | Multi-task loss in object detection          | Tasks re-weighting or modifying loss function | Task Weighting, Guided Loss        |
> > | Scale imbalance       | Objects with various scales and numbers      | Multi Scale features/images                   | Multi Scale CNN, FPN, NAS-FPN      |
> > | Spatial imbalance     | Different sizes, shapes, locations of boxes  | Cascade head and modifying regression loss    | Cascade R-CNN, Smooth L1/IoU loss  |
> > | Fg-Fg Class imbalance | Different objects' frequencies in nature     | Modifying sampling strategy, loss re-weighting| OFB sampling, RFS, Seesaw Loss     |
> > | Gradient variance imbalance    | **Effective batch size misalignment**        | Modulating gradient variance                  | AGVM                               |
> >
> > Even so, AGVM still works well for large-batch training under these imbalance problems. For example, Fg-Bg class imbalance, objective imbalance, scale imbalance, and spatial imbalance are naturally presented in COCO dataset and the results on COCO in this paper demonstrate the effectiveness of AGVM. For Fg-Fg class imbalance, we further conduct experiments with Mask R-CNN on LVIS (a long-tailed dense visual prediction benchmark) and report the Bbox mAP. We see AGVM still outperforms the baseline with large-batch setting even suffering from the long-tailed problem.
> >
> > | Batch size | Megdet  |  AGVM   |
> > |------------|---------|---------|
> > | 32         |21.4|21.4|
> > | 128        |20.8|21.3|
> > | 256        |20.1|20.7|
> > | 512        |NaN|19.9|
> >
> > [1] Oksuz, Kemal, et al. "Imbalance problems in object detection: A review." IEEE transactions on pattern analysis and machine intelligence 43.10 (2020): 3388-3415.

---

> ### Comment · Reviewer_KpQK · 2022-08-10
> **Acknowledgement**
>
> I would like to thank the authors for addressing my questions. Also, I appreciate my fellow reviewers' comments that lead to in-depth discussions with the authors.
>
> The authors well addressed my concerns. Specifically, the strategies that are introduced by the authors to improve numerical stability make sense to me.
>
> Additionally, I went over other reviewers' comments. As pointed out by reviewer 8EAD, the concerns are addressed at least to an acceptable level in my opinion.
>
> Therefore, I'd like to raise my rating to 6, i.e., weak accept.

---

> > ### Author Response · Authors · 2022-08-10
> > **Author Response**
> >
> > Dear Reviewer KpQK,
> >
> > We sincerely thank the reviewer for the constructive feedback and support!

---

### Official Review · Reviewer_8EAD · 2022-07-11

**Rating:** 7
**Confidence:** 4
**Soundness:** 3 good
**Presentation:** 3 good
**Contribution:** 3 good

**Summary:**

Authors study the problem of large-batch optimization for various "dense" computer vision tasks, such as object detection or instance segmentation. Their primary stated objective is to enable training these models with extremely large batches using multiple GPUs.
To achieve this objective, authors analyze the behavior of gradient variance and observe that the variance of different network sub-modules (e.g. detector backbone vs FPN) becomes dissimilar during training with large batch sizes. Based on this observation, authors propose Adaptive Gradient Variance Modulator (AGVM) - an optimizer-agnostic technique meant to balance the gradient variance between the sub-modules. Authors conduct experiments on several vision tasks and demonstrate that AGVM can scale to very large batch sizes.

**Questions:**



### Minor comments / typos:

> L159 they can be easily implemented using the popular deep learning platform e.g., PyTorch

Perhaps it would be better to paraphrase?
(A) **a / any** … platform **e.g.** PyTorch
(B) **the** … platform **i.e.** PyTorch

> Figure 1

The first row colors can be unduly associated with the second row colors. Would recommend using different color schemes.

>Table 3:

Why is batch 1536 in bold? (the caption indicates that bold denotes best results) If this is a deliberate formatting choice, I would recommend changing the caption or highlighting that batch size in a different manner.

> L192 Pytorch

Py**T**orch (consistency)

> The deliverables are released at https://anonymized-agvm.github.io/.

As of the first week of July, the above link only contains figures from the paper.
[Nit] While it does not affect my recommendation, i would still recommend to either use "deliverables **will be** released" or actually provide them during submission.



**Limitations:**

To the best of my understanding, the proposed problem (variance mismatch) and the solution (AGVM) appear more general than authors position them to be. As such, I believe it would be best to explain how is AGVM specific to dense visual predictions - or remove this limitation and evaluate it more generally. Furthermore, to the "extra overhead" could be addressed in more detail.
I elaborate on both these concerns above, in the "Strengths And Weaknesses" section.

**Strengths And Weaknesses:**

The paper proposes a practically useful solution to large-batch training with convincing practical experiments and hyperparameter sensitivity analysis.
By extending the applicability of large-batch training, authors not only allow training models in minutes on GPU clusters, but also make it feasible to train "dense prediction" tasks outside of compute clusters, such as in federated learning[1] or even volunteer computing[2], where large-batch training allows one to mitigate high communication latency. While i have many low-level concerns about the paper positioning and quality, I am strongly in favor of paper acceptance, since the proposed method could help "democratize" training in several computer vision problems. I list my concerns below.

### [conceptual] Dense vision tasks vs modular tasks

The paper positions AGVM as a solution to dense visual prediction tasks. However, the method itself does not seem specific to dense prediction, (and does necessarily work for all dense prediction tasks as authors admit in L319-320). Instead, AGVM appears to rely on the fact that a given task contains multiple sub-modules. Naturally, there are many "modular" tasks outside the area of dense prediction:
- NLP pre-training with ELECTRA[3] and derivative work
- reinforcement learning: DDPG[4] or SAC[5] (actor and critic networks)
- generative adversarial networks[6], (generator vs discriminator) cyclic GAN[7] contains 4 sub-modules

In short, there is a plethora of other tasks that appear to fit AGVM's motivation of sub-module variance.
Either the proposed AGVM fits those tasks - in which case, it is not specific to dense prediction - or there is some reason why it doesn't.
I believe that the paper would be more clearly positioned if authors explain why AGVM is specific to dense predition, or evaluate it on more general tasks.


### [practical] On measuring overhead

In Table 1 (and later), authors report "extra overhead" of less than 1%. In its current form, it is unclear what exactly does this overhead mean - and hence, predict how it will generalize to other hardware setups. At the very least, i would recommend clarifying the following issues:

- __communication per step:__ does AGVM double the required communication (MB / step) from the additional gradient all-reduce?
- __maximum memory:__ does AGVM need additional gradient buffers? (and hence, extra gpu memory in proportion to the model size)
- __measured quantity:__ does the overhead from Table 1 represent time, communication, flops, or energy overhead?

Ideally, it would be insightful to evaluate how the proposed overhead scales with network bandwidth and the number of GPUs.
If needed, the former can be emulated using `tc qdisc` on the active network interface.

[1] https://arxiv.org/pdf/1902.01046.pdf

[2] https://arxiv.org/pdf/2106.10207.pdf

[3] https://arxiv.org/pdf/2003.10555.pdf

[4] https://arxiv.org/pdf/1509.02971.pdf

[5] https://arxiv.org/pdf/1801.01290.pdf

[6] https://arxiv.org/pdf/1406.2661.pdf

[7] https://arxiv.org/pdf/1703.10593.pdf

---

> ### Author Response · Authors · 2022-08-02
> **Response to Reviewer 8EAD**
>
> Dear Reviewer 8EAD,
>
> Thank you for appreciating our approach. We will address your concerns below.
>
> **Q1: Dense vision tasks vs modular tasks**
>
> Thanks for this valuable comment.
> Specifically, AGVM is proposed to alleviate the large-batch training problem. Whether it works for a given task depends on two facts, one is that this given task contains multiple sub-modules such as the reviewer listed ELECTRA, DDPG, SAC, and GANs, and the other is that there is significant **effective batch size misalignment** between different sub-modules as we investigated in section 4.2, which is the essential reason for the inconsistent gradient variance. The feature pyramid network (FPN), region proposals, and the shared head between different levels are the core design ideas in dense visual prediction tasks. We find this inevitable introduces the **effective batch size misalignment**, leading to the inconsistent variance of different modules between GD and SGD. Based on this, AGVM is proposed to alleviate this inconsistency. After our verification, we find that the methods listed in the field of NLP, RL and GAN are not satisfied with the second fact above. Even so, we still agree with the reviewer's opinion that there will be other modular tasks satisfying both of the two facts. We will emphasize this in the revision and explore this in more research areas in future work.
>
> **Q2: On measuring overhead**
>
> Thanks for your constructive comment. Compared to the traditional data-parallel set-up, the only extra overhead in AGVM is the **additional all-reduce call time at t-th iteration when $t$%$\tau$=0**, and this overhead is highly related to the hardware set-up such as the cluster communication bandwidth.
> We will clarify the following issues in detail.
>
> **Communication per step**: AGVM doubles the required communication at t iteration when $t$%$\tau$=0.
>
> **Maximum memory**:  AGVM needs an additional gradient buffer to compute the cosine similarity in rank 0. In another word, AGVM needs an extra GPU memory to store the copy of $G_{t,2}^{i}$, which is in proportion to the model size. Take Faster R-CNN as an example, it needs an extra buffer to store these 42M gradients, but this is very small compared to the regular GPU memory used for forward and backward propagation.
>
> **Measured quality**: This overhead from Table 1 represents extra training time with AGVM compared with the basic setting, resulting in **0.12%** extra training time per epoch. The basic setting is 128 NVIDIA A100s with the regular data-parallel set-up,  Faster R-CNN+ResNet50 as the detector, 1024 batch size, and $\tau$=5. Even if we adopt batch size 2 per GPU (128 GPUs in total), the extra overhead is still negligible, e.g., less than 1%.
>
> **How the proposed overhead scales with network bandwidth and the number of GPUs**: The only extra overhead in AGVM is an additional all-reduce call time at t-th iteration when $t$%$\tau$=0. In our cluster, each computing node is connected via 8 InfiniBand 200Gb/s IB. The extra overhead of an all-reduce call for Faster R-CNN in our cluster is given as follows:
>
> | GPUs     | 16    | 32    | 64    | 128   | 256   | 512   |
> |----------|-------|-------|-------|-------|-------|-------|
> | Overhead | 3.1ms | 3.3ms | 3.7ms | 4.7ms | 5.6ms | 7.0ms |
>
> **Minors**: We have uploaded the rebuttal version to correct the minor typos.

---

> > ### Comment · Reviewer_8EAD · 2022-08-09
> > **Acknowledgement**
> >
> > I have read the response and appreciate the additional evaluations and discussion.
> > I have also read through the concerns raised by reviewrs 8KtA and q5yJ and the corresponding author responses.
> > For the practical concerns, I subjectively believe that authors have properly addressed these.
> > If reviewers 8KtA and q5yJ disagree, I would be eager to discuss that in the remaining time.
> >
> > Unfortunately, I cannot comment on theoretical issues in reviewer KpQK's position due to lack of expertise.

---

> > > ### Author Response · Authors · 2022-08-09
> > > **Thanks for appreciating our work!**
> > >
> > > We sincerely thank the reviewer for the constructive feedback and the kind support of this work! We believe AGVM helps democratize training in several computer vision problems. If our paper is accepted, we will definitely open source our codes.

---

### Author Response · Authors · 2022-08-02
**General Response: Contributions, New Experiments, and New Theoretical Analysis**

We sincerely appreciate all reviewers’ time and efforts in reviewing our paper. We are glad to find that reviewers generally recognized our contributions:

  * **Highlights.** The proposed method could help democratize training in several computer vision problems [8EAD].
  * **Theory.** The proposed method has a theoretical guarantee [8KtA].
  * **Experiments.** The proposed approach for large batch size training is simple and easy to implement [q5yJ]; Showing remarkable and promising results across different pipelines, optimizers, and datasets [8EAD, KpQK, 8KtA, q5yJ].
  * **Writing.** The paper is easy to follow and well-structured [8EAD, KpQK, 8KtA, q5yJ].


And we also thank all reviewers for their insightful and constructive suggestions, which help a lot in further improving our paper. In addition to the pointwise responses below, we summarize supporting experiments and theoretical analysis added in the rebuttal according to the reviewers’ suggestions.

**New Experiments:**
  * The proposed AGVM is more clearly positioned [8EAD];
  * Measuring overhead [8EAD];
  * Results on imbalanced object detection [KpQK];
  * Comparisons with Lena [8KtA];
  * Results for AdamW in Table 3 [8KtA];
  * Complete results for UniNet-G [8KtA];
  * The Same Variance-iteration Figure as the Figure 1 with AGVM [8KtA].
  * Experiments and clarification on the training with batch size 10k [q5yJ];

**New Theoretical Analysis:**
  * A rigorous guarantee for Eq (7) in the appendix (Eq. (12) in the rebuttal version) [8KtA];
  * Linear speedup property of AGVM [8KtA].

We hope our pointwise responses below could clarify all reviewers’ confusion and alleviate all concerns. We thank all reviewers’ time again. **We have uploaded the rebuttal version and the revised parts are highlighted in red.**

---

### Meta-Review · Area_Chair_gZiq · 2022-08-26

**Recommendation:** Accept
**Confidence:** Less certain

**Metareview:**

The authors describe a new method of large-batch optimisation for dense prediction computer vision tasks. The reviewers appreciate the simplicity of the method, convincing experiments and the potential practical importance. AC recommends acceptance.

**Award:**

No

---

### Decision · Program_Chairs · 2022-09-14

Accept